# SELF: A Robust Singular Value and Eigenvalue Approach for LLM Fingerprinting

## Abstract

The protection of Intellectual Property (IP) in Large Language Models (LLMs) represents a critical challenge in contemporary AI research. While fingerprinting techniques have emerged as a fundamental mechanism for detecting unauthorized model usage, existing methods—whether behavior-based or structural—suffer from vulnerabilities such as false claim attacks or susceptible to weight manipulations. To overcome these limitations, we propose SELF, a novel intrinsic weight-based fingerprinting scheme that eliminates dependency on input and inherently resists false claims. SELF achieves robust IP protection through two key innovations: 1) unique, scalable and transformation-invariant fingerprint extraction via singular value and eigenvalue decomposition of LLM attention weights, and 2) effective neural network-based fingerprint similarity comparison based on few-shot learning and data augmentation. Experimental results demonstrate SELF maintains high IP infringement detection accuracy while showing strong robustness against various downstream modifications, including quantization, pruning, and fine-tuning attacks. Our code is available at `https://anonymous.4open.science/r/SELF-BC5F`.

## 1 Introduction

Large language models (LLMs) are increasingly being adopted as versatile tools to enhance productivity in various fields, including medical assistance (Thirunavukarasu et al. (2023)), code generation (Fakhoury et al. (2024)), and so on. Developing a functional LLM requires substantial investments, including high-quality datasets, significant computational resources, and specialized human expertise. Consequently, protecting the intellectual property (IP) of LLMs is of paramount importance (Zheng et al. (2024)), particularly in the current era where open-source trends clash with the need for model creators to maintain naming conventions for attribution on derivative works.

Current model IP infringement detection methods primarily fall into two categories: watermarking and fingerprinting. Watermarking approaches embed identifiable features (watermarks) invasively into target models while trying to preserve their original functionality (Zhang et al. (2024); Xu et al. (2024b)). In contrast, fingerprinting methods extract unique model identifiers without modifying the model, either by analyzing the model's input-output behavioral patterns (Gubri et al. (2024)) (i.e., behavior fingerprinting) or structural information (i.e., structural fingerprinting) such as weight distributions (Zeng et al. (2024)), intermediate representations (Zhang et al. (2025)), or gradient profiles (Wu et al. (2025)). Compared to watermarking-based methods, fingerprinting schemes eliminate the need of retraining and avoid potential performance degradation associated with watermark insertion (Zheng et al. (2022)).

Despite these advantages, existing fingerprinting methods face critical limitations. Behavior-based techniques are vulnerable to false claim attacks (Liu et al. (2024b)), wherein malicious actors can falsely claim the ownership of independently trained models by crafting (transferable) adversarial samples. Although Shao et al. (2025) propose to mitigate the attack by constructing fingerprints using targeted adversarial examples, the risk persists as such adversarial examples can still be transferrable albeit with greater difficulty. Structural approaches analyze model internal parameters but lack robustness against weight manipulations such as permutation or linear mapping. For schemes like HuRef (Zeng et al. (2024)) where the input is required to actively participate in fingerprint computation, we further extend the scope of false claim attack as malicious accuser can manipulate

ownership verification results through carefully crafted input. Under this broader definition, we conducted false claim attack on HuRef scheme and successfully manipulated the similarity score output (see Appendix B).

To address these issues, we propose a structural fingerprinting method named SELF, which purely depends on the model weights. Figure 1 describes SELF's pipeline. The owner first extracts a fingerprint from the target model and trains a Similarity Network (*SimNet*) for verification. If the model is stolen, the owner can detect piracy by *SimNet*'s high similarity output. SELF comprises two key components: (1) *Fingerprint Extraction*, which derives unique, robust and scalable fingerprints from model weights; and (2) *Similarity Computation*, where a neural network learns fingerprint patterns to enable robust and efficient similarity assessment.

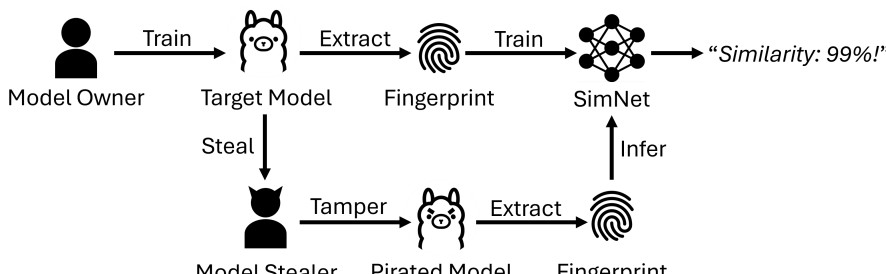

Figure 1: IP infringement detection pipeline using SELF.

In the fingerprint extraction module, we address potential model weight tampering caused by transformation attacks (e.g., permutation and linear-mapping (Zeng et al. (2024))) through identifying invariant attributes. Specifically, we first compute singular value and eigenvalue invariant matrices from the weight matrices, then derive their corresponding invariant singular values and eigenvalues to construct the model fingerprint. Leveraging the inherent properties of these values, the fingerprint remains both invariant against transformation attacks and robust to downstream modifications such as quantization and fine-tuning.

The similarity computation is performed by a *SimNet*, trained in a few-shot learning paradigm with a limited set of related and unrelated model fingerprints. To enhance generalization, we employ data augmentation techniques including noise injection, row/column deletion, and element masking on the training dataset. This approach ensures robust and accurate similarity comparisons under constrained sample conditions.

Our contributions can be summarized as follows:

1. We present SELF, a weight-based fingerprinting method for LLMs that is inherently resistant to false claim attack. This weight-exclusive methodology eliminates dependency on input samples, thereby preventing false claim of ownership caused by manipulating adversarial inputs.

2. We propose robust fingerprint extraction against transformation attacks. By deriving fingerprints from singular values and eigenvalues of matrices constructed from model weights, our approach leverages their fundamental mathematical properties to ensure uniqueness, scalability, and robustness.

3. Our approach employs a *SimNet* to learn distinctive patterns from the extracted fingerprints. To address data scarcity in the few-shot learning scenario, we implement data augmentation strategies, which enable robust and effective identification of IP infringement.

4. Our comprehensive evaluation across Qwen2.5-7B, Llama2-7B, and their related / unrelated models demonstrates that the proposed method achieves both accurate and robust IP infringement detection. The method maintains its discriminative capability even when subjected to various model modifications, including quantization, pruning, and fine-tuning attacks.

## 2 PRELIMINARIES

### 2.1 ATTENTION MECHANISM

LLMs widely adopt the Transformer architecture (Vaswani et al. (2017)), with the attention mechanism serving as its core component. The computation of this mechanism can be formulated as:

$$H_{\text{out}} = \text{softmax}\left(\frac{H_{\text{in}}W_Q(H_{\text{in}}W_K)^{\top}}{\sqrt{d}}\right)(H_{\text{in}}W_V)W_O \tag{1}$$

where $H_{\text{in}}, H_{\text{out}} \in \mathbb{R}^{n \times d_{\text{model}}}$ denote the input hidden representations and the self-attention output, respectively, $W_Q, W_K, W_V \in \mathbb{R}^{d_{\text{model}} \times d}$ represent the learnable weight matrices for the Query, Key, and Value matrices, respectively, while $W_O \in \mathbb{R}^{d \times d_{\text{model}}}$ denotes the output weight matrix. Here, $n$, $d$ and $d_{\text{model}}$ correspond to sequence length, projection dimension, and embedding dimension, respectively.

### 2.2 TRANSFORMATION ATTACKS

Model weights are a direct consequence of the specific learning process the model goes through and the data it was trained on, therefore can serve as unique fingerprints for model similarity comparison. However, weights-based fingerprint verification may be evaded by the following transformation attacks, which aim to preserve the model's function behavior but significantly modify the weights (Zeng et al. (2024)):

**Permutation Attack** This attack rearranges the weights by permuting the weight matrix. Let $P \in \mathbb{R}^{d_{\text{model}} \times d_{\text{model}}}$ denote an arbitrary permutation matrix. This permutation transformation needs to be applied to the input of the model and corresponding weights:

$$\begin{aligned}
\hat{H}_{\text{in}} &= H_{\text{in}}P^T, \\
\hat{W}_Q &= PW_Q, \quad \hat{W}_K = PW_K, \quad \hat{W}_V = PW_V, \quad \hat{W}_O = W_O P^T
\end{aligned} \tag{2}$$

As a result, $\hat{H}_{\text{out}} = H_{\text{out}}P^T$. The permutation effect propagates through subsequent layers, making each layer's input and output be permuted with $P^T$. The original model function can be preserved by applying the same permutation $P$ to the final output.

**Linear Mapping Attack** Alternatively, a pair of linear transformations can be applied to the attention weights:

$$\hat{W}_Q = W_Q C_1, \quad \hat{W}_K = W_K C_1^{-T}, \quad \hat{W}_V = W_V C_2, \quad \hat{W}_O = C_2^{-1} W_O \tag{3}$$

where $C_1, C_2 \in \mathbb{R}^{d \times d}$ are arbitrary invertible matrices and $T$ denotes the transpose. This linear transformation preserves the attention scores, e.g., $(H_{\text{in}}\hat{W}_Q)(H_{\text{in}}\hat{W}_K)^T = H_{in}W_Q C_1 C_1^{-1} W_K^T H_{in}^T = (H_{\text{in}}W_Q)(H_{\text{in}}W_K)^T$, $(H_{\text{in}}\hat{W}_V)\hat{W}_O = (H_{\text{in}}W_V C_2)C_2^{-1}W_O = (H_{\text{in}}W_V)W_O$, leaving the output $H_{\text{out}}$ unchanged.

The two attacks highlight a fundamental vulnerability: the model output remains unaffected, while the model weights are altered in ways that invert similarity metrics computed purely on weight comparison.

## 3 EXPLORING SINGULAR VALUES AND EIGENVALUES AS INVARIANT FINGERPRINTS

Singular values and eigenvalues are fundamental matrix attributes that capture intrinsic properties of a matrix. This section introduces the calculation of the two features and discusses their invariance to transformation attacks.

### 3.1 SINGULAR VALUES

Singular values are defined for matrix with any size and describe how much the matrix stretches space along principal mutually orthogonal directions. Any complex matrix $M \in \mathbb{C}^{m \times n}$ admits a singular value decomposition (SVD):

$$M = U\Sigma V^*, \tag{4}$$

where $U \in \mathbb{C}^{m \times m}$ and $V \in \mathbb{C}^{n \times n}$ are unitary matrices. The matrix $\Sigma \in \mathbb{R}^{m \times n}$ is a rectangular diagonal matrix with non-negative real numbers $\sigma_1 \geq \sigma_2 \geq \cdots \geq 0$ on the diagonal. These values $\sigma_i = \Sigma_{ii}$ for $i = 1, 2, \ldots, \min\{m, n\}$ are uniquely determined by $M$ and are referred to as the *singular values* of $M$.

**Singular values remain unchanged under row, column permutation or any combination of both.** Specifically, let $P_1 \in \mathbb{R}^{m \times m}$, $P_2 \in \mathbb{R}^{n \times n}$ be the row and column permutation matrices, respectively, and $\sigma_i(\cdot)$ denote the $i_{th}$ singular value of the corresponding matrix. Since permutation matrices are orthogonal, we have:

$$\sigma_i(M) = \sigma_i(P_1 M) = \sigma_i(M P_2) = \sigma_i(P_1 M P_2). \tag{5}$$

This property (Franklin (2000)) enables the use of singular values as robust descriptors for model weight matrices, providing resistance to the permutation attacks defined in equation 2 through the construction of singular value invariant matrix in Section 4.2.

**Singular values exhibit stability under small perturbations.** Let $\Delta M$ denote the perturbations added to matrix $M$, according to Weyl's inequality (Franklin (2000)):

$$|\sigma_i(M + \Delta M) - \sigma_i(M)| \leq \|\Delta M\|_2, \tag{6}$$

where $\| \cdot \|_2$ denotes the spectral norm. This implies that the singular values of a matrix remain approximately unchanged as long as the perturbation magnitude $\|\Delta M\|_2$ is small.

### 3.2 EIGENVALUES

Eigenvalues are defined only for square matrices and describe the scaling factors associated with specific invariant directions (eigenvectors). For a square matrix $N \in \mathbb{C}^{n \times n}$, the eigenvalues $\lambda_1, \ldots, \lambda_n$ are defined as the roots of its characteristic polynomial, which are obtained by solving:

$$\det(N - \lambda I) = 0, \tag{7}$$

where $\det$ denotes determinant and $I$ is the identity matrix. If $N$ is diagonalizable, eigenvalues can be more efficiently and stably computed via the following eigenvalue decomposition (EVD):

$$N = Q\Lambda Q^{-1}, \tag{8}$$

where $Q \in \mathbb{C}^{n \times n}$ consists of the linearly independent eigenvectors of $N$, and $\Lambda \in \mathbb{C}^{n \times n}$ is a diagonal matrix whose diagonal entries $\Lambda_{ii} = \lambda_i$ for $i = 1, 2, \ldots, n$ are the eigenvalues of $N$.

**Eigenvalues are invariant under similarity transformations.** Let $C$ be an invertible matrix, the operation of transforming a matrix $N$ to $\hat{N} = CNC^{-1}$ is called a similarity transformation. The eigenvalues of $\hat{N}$ can be obtained by solving the roots of

$$det(\hat{N} - \lambda I) = det(CNC^{-1} - \lambda I) = det(CNC^{-1} - \lambda I(CC^{-1}) = det(CNC^{-1} - C(\lambda I)C^{-1})$$
$$= det(C(N - \lambda I)C^{-1}) = det(C)det(N - \lambda I)det(C^{-1})$$
$$= det(C)det(N - \lambda I)\frac{1}{det(C)} = det(N - \lambda I). \tag{9}$$

Therefore, $\hat{N}$ and $N$ share the same eigenvalues. This property forms the theoretical foundation for defending against the linear mapping attack described in equation 3 through the construction of eigenvalue invariant matrix in Section 4.2.

**Eigenvalues are robust against small perturbations.** The Bauer-Fike theorem (Bauer & Fike (1960)) provides a quantitative bound on the sensitivity of eigenvalues under perturbation. Let $\Delta N$ denote the perturbation on $N$ and $\lambda_i(\cdot)$ denote the $i_{th}$ eigenvalue of the corresponding matrix. If $N$ is diagonalizable, then for any perturbed matrix $N + \Delta N$ and any $\lambda_i(N + \Delta N)$, there always exists a $\lambda_j(N)$ satisfies:

$$|\lambda_i(N + \Delta N) - \lambda_j(N)| \le \kappa(Q) \cdot \|\Delta N\|_2, \tag{10}$$

where $\kappa(Q) = \|Q\|_2 \|Q^{-1}\|_2$ is the condition number of $Q$. Thus, if $\Delta N$ is small in spectral norm and $\kappa(Q)$ is not too large, the eigenvalues of the perturbed matrix can be close to those of the original. Moreover, a very large $\kappa(Q)$ indicates a high nonnormal $N$, which can undermine the reliability and convergence of numerical methods (Chaitin-Chatelin (1997)), and thus should be avoided in deep learning systems.

In the context of model fingerprinting, both singular values and eigenvalues can serve as transformation-invariant descriptors, providing robustness against adversarial manipulations of model weights.

## 4 METHODOLOGY

Our method consists of two components: 1) Fingerprint Extraction. This component analyzes the model weights to extract unique and scalable fingerprints by leveraging the properties of singular values and eigenvalues. 2) Similarity Computation. We employ a neural network to learn patterns from the extracted fingerprints, enabling efficient and robust model similarity assessment. The following sections introduce the threat model and these two essential components.

### 4.1 THREAT MODEL

Our threat model mainly involves three roles: the model owner (original developer or authorized distributor), the model stealer (adversary), and the judger (the trusted third party). The model owner owns the IP of the **target model** (the model to be protected) and maintains white-box access to the model. The model stealer pirates (via extraction, insider theft, or open-source leaks, etc.) the target model and may modify (via fine-tuning, pruning, quantization, or weight transformation attacks, etc.) it to evade IP detection while preserving model functionality. Once the model owner flags a **suspect model** (suspicious to be pirated), he/she could request IP verification procedures by the judger. The judger temporarily obtains white-box access to both the target and suspect models only during the investigation, and its objective is to reliably verify model provenance despite potential adversarial modifications. Crucially, the mechanism must distinguish between: **related models** (derived from the target, even if modified) and **unrelated models** (independently trained), while operating without knowledge of the specific attack employed. Our method aims to provide accurate, robust, and efficient infringement detection for IP forensic scenarios.

### 4.2 FINGERPRINT EXTRACTION VIA MATRIX DECOMPOSITION

Figure 2 shows the fingerprint extraction pipeline. Given an LLM model, we extract its fingerprint by analyzing the first $N_{\mathcal{F}}$ Transformer block layers. For each layer $i$, we compute a layer-wise fingerprint $\mathcal{F}^i$. The overall fingerprint $\mathcal{F}$ of the model can be obtained by aggregating all $N_{\mathcal{F}}$ layer-wise fingerprints.

To extract robust fingerprint, we focus on invariant features derived from the core component of the LLM, i.e., the attention blocks. Compared to other components, the advantage of the attention block lies in (1) High-dimensionality: Attention matrices encodes rich, high-dimensional features (e.g.,$W_Q$, $W_K$ account for >30% of Llama2-7B's parameters), while LayerNorm has a much smaller parameter size (e.g., 0.1% of Llama2-7B's total parameters) and outputs vectors, which limit its utility for SVD- and EVD-based fingerprint extraction. (2) Architectural generality: Attention blocks are universal across transformer variants, while other blocks such as MLPs (e.g., in Mixture of Experts models (Shazeer et al. (2017))) exhibit structural variability that could undermines consistency.

Let $W_Q \in \mathbb{R}^{d_{\text{model}} \times d}$, $W_K \in \mathbb{R}^{d_{\text{model}} \times d}$, $W_V \in \mathbb{R}^{d_{\text{model}} \times d}$ and $W_O \in \mathbb{R}^{d \times d_{\text{model}}}$ denote the query, key, value and output projection weight matrices in a Transformer attention block, where $d_{\text{model}}$ is

the embedding dimension and $d$ is the projection dimension. As described in Section 2.2, these matrices may be subjected to the permutation and linear mapping attacks of the following form:

$$\hat{W}_Q = PW_QC_1, \quad \hat{W}_K = PW_KC_1^{-T}, \quad \hat{W}_V = PW_VC_2, \quad \hat{W}_O = C_2^{-1}W_OP^T \quad (11)$$

where $P \in \mathbb{R}^{d_{\text{model}} \times d_{\text{model}}}$ is a permutation matrix, and $C_1, C_2 \in \mathbb{R}^{d \times d}$ are invertible matrices. To counter these attacks, we introduce two transformation-invariant matrices for robust fingerprinting.

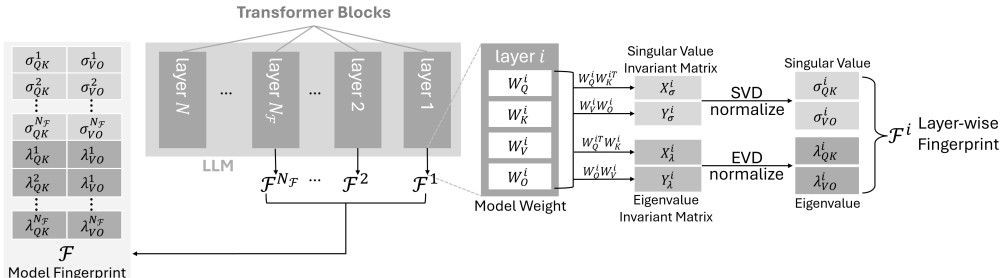

Figure 2: Fingerprint extraction via matrix decomposition. For a given model, we extract its fingerprint using the first $N_\mathcal{F}$ Transformer block layers. Specifically, we first compute the fingerprint of each individual layer $\mathcal{F}^i$ and then aggregate the $N_\mathcal{F}$ layer-wise fingerprints to form the overall model fingerprint $\mathcal{F}$. For each layer, we first compute the singular value invariant matrices $X_\sigma^i, Y_\sigma^i$ and the eigenvalue invariant matrices $X_\lambda^i, Y_\lambda^i$ from the attention weights $W_Q^i, W_K^i, W_V^i$, and $W_O^i$. Then, we extract the normalized singular value vector $\sigma_{QK}^i, \sigma_{VO}^i$ and eigenvalue vector $\lambda_{QK}^i, \lambda_{VO}^i$, which together form the fingerprint of layer $i$.

**Singular Value Invariant Matrix** We first define matrices $X_\sigma$ and $Y_\sigma$ that preserves singular values under the transformation attacks:

$$X_\sigma = W_QW_K^T \in \mathbb{R}^{d_{\text{model}} \times d_{\text{model}}}$$
$$Y_\sigma = W_VW_O \in \mathbb{R}^{d_{\text{model}} \times d_{\text{model}}} \quad (12)$$

**Theorem 1** (Singular Value Invariance). *Under the transformation attack described in equation 11, the matrices $\hat{X}_\sigma = \hat{W}_Q\hat{W}_K^T$ and $\hat{Y}_\sigma = \hat{W}_V\hat{W}_O$ satisfies:*

$$\hat{X}_\sigma = (PW_QC_1)(PW_KC_1^{-T})^T = PW_QC_1C_1^{-1}W_K^TP^T = PW_QW_K^TP^T = PX_\sigma P^T$$
$$\hat{Y}_\sigma = (PW_VC_2)(C_2^{-1}W_OP^T) = PW_VC_2C_2^{-1}W_OP^T = PW_VW_OP^T = PY_\sigma P^T \quad (13)$$

*Since permutation matrices are orthogonal, $X_\sigma$ and $\hat{X}_\sigma$, $Y_\sigma$ and $\hat{Y}_\sigma$ are orthogonally equivalent and consequently share the same singular values.*

**Eigenvalue Invariant Matrix** Next, we define matrices $X_\lambda$ and $Y_\lambda$ that preserves eigenvalues under the transformation attack:

$$X_\lambda = W_Q^TW_K \in \mathbb{R}^{d \times d}$$
$$Y_\lambda = W_OW_V \in \mathbb{R}^{d \times d} \quad (14)$$

**Theorem 2** (Eigenvalue Invariance). *Under the transformation attack described in equation 11, the matrix $\hat{X}_\lambda = \hat{W}_Q^T\hat{W}_K$ and $\hat{Y}_\lambda = \hat{W}_O\hat{W}_V$ satisfies:*

$$\hat{X}_\lambda = (PW_QC_1)^T(PW_KC_1^{-T}) = C_1^TW_Q^TP^TPW_KC_1^{-T} = C_1^TW_Q^TW_KC_1^{-T} = C_1^TX_\lambda C_1^{-T}$$
$$\hat{Y}_\lambda = (C_2^{-1}W_OP^T)(PW_VC_2) = C_2^{-1}W_OP^TPW_VC_2 = C_2^{-1}W_OW_VC_2 = C_2^{-1}Y_\lambda C_2 \quad (15)$$

*This establishes a similarity transformation between $X_\lambda$ and $\hat{X}_\lambda$, $Y_\lambda$ and $\hat{Y}_\lambda$, guaranteeing each matrix pair share identical eigenvalues.*

**Fingerprint for a Single Layer:** Let $W_Q^i$, $W_K^i$, $W_V^i$ and $W_O^i$ denote the query, key, value, and output matrices of the $i_{th}$ Transformer block layer. We compute the singular value invariant matrices $X_\sigma^i$ and $Y_\sigma^i$ as defined in equation 12 and the eigenvalue invariant matrices $X_\lambda^i$ and $Y_\lambda^i$ as defined in equation 14. From the two matrices, we extract the following features sorted in descending order by magnitude:

1. the top $h$ singular values of $X_\sigma^i$, denoted by vector $\tilde{\sigma}_{QK}^i \in \mathbb{R}^h$,

2. the top $h$ eigenvalues (by magnitude) of $X_\lambda^i$, denoted by vector $\tilde{\lambda}_{QK}^i \in \mathbb{R}^h$,

3. the top $h$ singular values of $Y_\sigma^i$, denoted by vector $\tilde{\sigma}_{VO}^i \in \mathbb{R}^h$,

4. the top $h$ eigenvalues (by magnitude) of $Y_\lambda^i$, denoted by vector $\tilde{\lambda}_{VO}^i \in \mathbb{R}^h$,

then each vector is normalized to unit $L_2$ norm, e.g., $\sigma_{QK}^i = \frac{\tilde{\sigma}_{QK}^i}{\|\tilde{\sigma}_{QK}^i\|_2}$, and the fingerprint for layer $i$ is then defined with the normalized vectors as $\mathcal{F}^i = [\sigma_{QK}^i, \lambda_{QK}^i, \sigma_{VO}^i, \lambda_{VO}^i]^T \in \mathbb{R}^{4 \times h}$.

**Fingerprint for the Whole Model:** As studied by Zheng et al. (2022), lower-layer (near to the input) weights tend to be more robust against fine-tuning or task-specific adaptations. Therefore, we extract fingerprints from the first $N_\mathcal{F}$ Transformer layers. The full model fingerprint is obtained by stacking all singular and eigenvalue vectors in the following order:

$$\mathcal{F} = [\sigma_{QK}^1, \sigma_{QK}^2, \ldots, \sigma_{QK}^{N_\mathcal{F}}, \lambda_{QK}^1, \lambda_{QK}^2, \ldots, \lambda_{QK}^{N_\mathcal{F}},$$
$$\sigma_{VO}^1, \sigma_{VO}^2, \ldots, \sigma_{VO}^{N_\mathcal{F}}, \lambda_{VO}^1, \lambda_{VO}^2, \ldots, \lambda_{VO}^{N_\mathcal{F}}]^T \in \mathbb{R}^{4N_\mathcal{F} \times h} \tag{16}$$

This fingerprint matrix $\mathcal{F}$ captures intrinsic structural information from multiple attention layers, while remaining robust to the weight transformation attacks.

### 4.3 FINGERPRINT SIMILARITY COMPUTATION VIA NEURAL NETWORK

To determine whether a suspect model is a derived variant of the target model, we adopt a residual network (He et al. (2016)) based fingerprint similarity network, denoted as *SimNet* : $\mathbb{R}^{4N_\mathcal{F} \times h} \to [0, 1]$, which maps a model fingerprint $\mathcal{F} \in \mathbb{R}^{4N_\mathcal{F} \times h}$ to a similarity score ranges from 0 to 1.

To train *SimNet*, we firstly construct a labeled training set: $\mathcal{D}_{\text{train}} = \{(\mathcal{F}_i, y_i)\}_{i=1}^n$, where $y_i = 1$ if $\mathcal{F}_i$ originates from the target model or a related model, and $y_i = 0$ if it is from an unrelated model.

Due to limited training samples, we train *SimNet* using a few-shot learning scheme, employing data augmentation to expand $\mathcal{D}_{\text{train}}$. Specifically, we augment $\mathcal{D}_{\text{train}}$ by applying Gaussain noise, row deletion, column deletion, and random masking to $W_Q$, $W_K$, $W_V$ and $W_O$ of the first $N_\mathcal{F}$ Transformer layers.

Specifically, we modify the attention weights in the first $N_\mathcal{F}$ Transformer layers as follows:

1. **Gaussian Noise:** Add random Gaussian noise sampled i.i.d. from the standard normal distribution–$\mathbf{N}_Q, \mathbf{N}_K, \mathbf{N}_V, \mathbf{N}_O \sim \mathcal{N}(0, 1)$– to the weight matrices with strength $\alpha$:

$$\tilde{W}_Q = W_Q + \alpha\mathbf{N}_Q, \quad \tilde{W}_K = W_K + \alpha\mathbf{N}_K,$$
$$\tilde{W}_V = W_V + \alpha\mathbf{N}_V, \quad \tilde{W}_O = W_O + \alpha\mathbf{N}_O. \tag{17}$$

2. **Row Deletion:** Randomly select a subset of indices $\mathcal{I}_r \subset \{1, \ldots, d_{\text{model}}\}$ with $|\mathcal{I}_r| = n_r$, and delete the corresponding rows from $W_Q$, $W_K$, and $W_V$, and delete the corresponding columns from $W_O$.

3. **Column Deletion:** Randomly select a subset of indices $\mathcal{I}_c \subset \{1, \ldots, d\}$ with $|\mathcal{I}_c| = n_c$, and delete the corresponding columns from $W_Q$, $W_K$, and $W_V$, and delete the corresponding rows from $W_O$.

4. **Random Masking:** Mask the weight matrices based on a predefined threshold $r$ and random noise matrix sampled i.i.d from a uniform distribution– $\mathbf{N}_Q, \mathbf{N}_K, \mathbf{N}_V, \mathbf{N}_O \sim$

$\mathcal{U}(0,1)^{d_{\text{model}} \times d}$:

$$\tilde{W}_Q(i,j) = W_Q(i,j) \odot \mathbb{M}[\mathbf{N}_Q(i,j) \geq r],$$
$$\tilde{W}_K(i,j) = W_K(i,j) \odot \mathbb{M}[\mathbf{N}_K(i,j) \geq r],$$
$$\tilde{W}_V(i,j) = W_V(i,j) \odot \mathbb{M}[\mathbf{N}_V(i,j) \geq r],$$
$$\tilde{W}_O(i,j) = W_O(i,j) \odot \mathbb{M}[\mathbf{N}_O^T(i,j) \geq r].$$

(18)

where $\odot$ is the element-wise multiplication and $\mathbb{M}[\mathbf{N} > r]$ is a binary matrix where each element is 1 if $N_{i,j} \geq r$, and 0 otherwise.

The augmented weights $(\tilde{W}_Q, \tilde{W}_K, \tilde{W}_V, \tilde{W}_O)$ are used to compute the invariant matrices $\tilde{X}_\sigma, \tilde{X}_\lambda, \tilde{Y}_\sigma$, and $\tilde{Y}_\lambda$, from which we derive the augmented fingerprint $\tilde{\mathcal{F}}$. The final training dataset is $\tilde{\mathcal{D}}_{\text{train}} = \mathcal{D}_{\text{train}} \cup \left\{ (\tilde{\mathcal{F}}_j, y_j) \right\}_{j=1}^{\tilde{n}}$, where $\tilde{\mathcal{F}}_j$ denotes an augmented fingerprint generated from a model with label $y_j$, and $\tilde{n}$ is the number of augmented fingerprint samples.

With the augmented dataset $\tilde{\mathcal{D}}_{\text{train}}$, *SimNet* for the target model $T$ can be trained. Given a suspect model $S$ and a predefined similarity threshold $\tau \in [0,1]$, if $SimNet(\mathcal{F}_S) > \tau$, $S$ is considered potentially related to the target model due to the high similarity score. Otherwise, the suspect model is detected as unrelated models.

## 5 EXPERIMENTS

**Experiment Settings:** we extract fingerprints from the first $N_{\mathcal{F}} = 8$ layers, using the top $h = 256$ singular values and eigenvalues, resulting in fingerprint size of $32 \times 256$. For each target model, the training dataset for *SimNet* includes the extracted fingerprints of the model itself, one of its fine-tuned offspring and three unrelated models. Model architecture and training details of *SimNet* can be found in Appendix E.

### 5.1 EFFECTIVENESS VERIFICATION

We use Qwen2.5-7B and Llama2-7B as target models to evaluate the effectiveness of SELF, and their publicly available offsprings are used as related models. For unrelated models, we select 10 independent open-source models with diverse architectures and parameter sizes ranging from 800M to 7B. As shown in Table 1, the similarity score of related models are greater than 0.9 while those for unrelated models are less than 0.3, demonstrating that our method can effectively discriminate between related and unrelated models.

Table 1: Fingerprint Similarity (Target Models: Qwen2.5-7B and Llama2-7B)

| Qwen2.5-7B Unrelated | | Qwen2.5-7B Related | | Llama2-7B Unrelated | | Llama2-7B Related | |
|---|---|---|---|---|---|---|---|
| **Model** | **Score** | **Model** | **Score** | **Model** | **Score** | **Model** | **Score** |
| Mistral-7B-V0.3 | 0.0050 | **Fine-tuned Variants** | | Mistral-7B-V0.3 | 0.0050 | **Fine-tuned Variants** | |
| Llama2-7B | 0.0020 | Qwen2.5-7B-Instruct | 0.9950 | Qwen1.5-7B | 0.2902 | Llama-2-7B-Chat | 0.9950 |
| Baichuan2-7B | 0.0009 | Qwen2.5-Math-7B | 0.9979 | Baichuan2-7B | 0.1862 | CodeLlama-7B | 0.9641 |
| InternLM2.5-7B | 0.0139 | Qwen2.5-Coder-7B | 0.9910 | InternLM2.5-7B | 0.0050 | Llemma-7B | 0.9699 |
| GPT2-Large | 0.0050 | TableGPT-7B | 0.9944 | GPT2-Large | 0.2372 | **Pruned Variants** | |
| Cerebras-GPT-1.3B | 0.0011 | Qwen2.5-7B-Medicine | 0.9950 | Cerebras-GPT-1.3B | 0.2874 | Sheared-Llama-2.7B | 0.9917 |
| ChatGLM2-6B | 0.0091 | Qwen2.5-7B-abliterated-v2 | 0.9950 | ChatGLM2-6B | 0.0057 | SparseLlama-2-7B | 0.9941 |
| OPT-6.7B | 0.0005 | **Quantized Variants** | | OPT-6.7B | 0.0009 | **Quantized Variants** | |
| Pythia-6.9B | 0.0156 | Qwen2.5-7B-4bit | 0.9950 | Pythia-6.9B | 0.0002 | Llama2-7B-4bit | 0.9950 |
| MPT-7B | 0.0050 | Qwen2.5-7B-8bit | 0.9950 | MPT-7B | 0.0050 | Llama2-7B-8bit | 0.9950 |

High similarity ($>0.5$) means potential IP infringement.

It is important to note that Qwen2.5-7B employs Grouped-Query Attention (GQA), which groups multiple query heads to share a single key-value head pair. This architecture introduces a dimensionality mismatch with previous definition in Sec. 2.1, as the number of key and value heads is smaller than that of the query heads. To resolve this compatibility issue, we conceptually broadcast the shared weights in $W_K$ and $W_V$ to align with the number of query heads. This replication functions as a structured weight-sharing constraint, ensuring GQA-based models like Qwen2.5-7B to be seamlessly integrated into our fingerprinting framework.

To provide intuitive performance of the fingerprint $\mathcal{F}$'s discriminability, we further include in the Appendix D the fingerprint distance between the above models and three DeepSeek-R1 (DeepSeek-AI et al. (2025)) distilled models versus their base models, demonstrating effectiveness across diverse architectures.

## 5.2 ROBUSTNESS VERIFICATION

We assess the robustness of SELF by evaluating its resilience against carefully crafted fine-tuning attacks and model pruning. A robust fingerprint should maintain a high similarity score while preserving model performance under such modification.

### 5.2.1 ROBUSTNESS AGAINST FINE-TUNING ATTACK

An attacker may fine-tune the stolen model to evade IP detection by intentionally adjusting model parameters so that the extracted fingerprint $\mathcal{F}_S$ is deviated from the original $\mathcal{F}_T$. Formally, this is achieved by optimizing the attack loss $L_{attack} = 1/(\|\mathcal{F}_M - \mathcal{F}_T\|_2 + \epsilon)$, where $\mathcal{F}_M$ is the fingerprint of the stolen model and $\epsilon = 10^{-9}$ serves as a small constant for numerical stability. Moreover, an attacker may also try to use actual dataset to maintain the model's performance, optimizing a combined objective incorporating both the attack loss and the dataset loss, i.e., $l_1 L_{attack} + l_2 L_{data}$. In the experiment, we use the cross-entropy loss on WikiText2 dataset as $L_{data}$. To keep the two loss functions on the same scale, we adopt $l_1 = 0.1, l_2 = 1$.

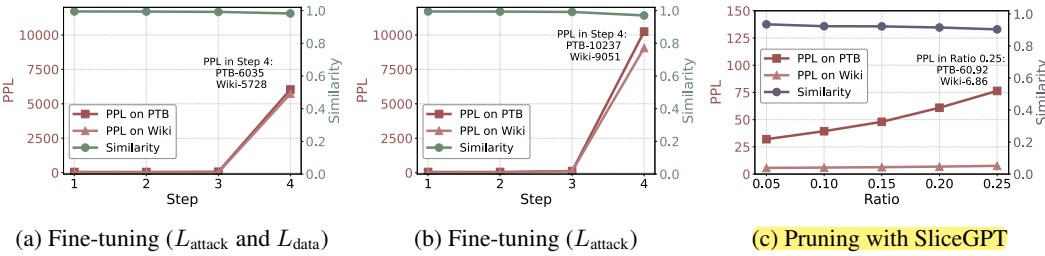

(a) Fine-tuning ($L_{attack}$ and $L_{data}$)    (b) Fine-tuning ($L_{attack}$)    (c) Pruning with SliceGPT

Figure 3: Fingerprint similarity and PPL change of Llama2-7B under different attacks. (a) and (b) show the results under fine-tuning attacks. (c) shows the results under SliceGPT pruning. Since State-of-The-Art LSTM achieves PPL smaller than 60 on PTB dataset (Merity et al. (2018)), we consider a PPL of 60 or higher as "unacceptable" for larger models like transformer.

We implemented this fine-tuning attack on the Llama2-7B model, employing a learning rate of $5 \times 10^{-3}$ and evaluating the model's performance change. To account for randomness, we averaged the results from four random seeds (10, 42, 99, 1024). We analyze both fingerprint similarity and model performance degradation, measuring the latter using perplexity (PPL) on WikiText2 (Merity et al. (2016)) and PTB (Marcus et al. (1993)) benchmarks. A lower PPL indicates better performance. Figures 3a and 3b show that even if the target model has been extensively fine-tuned to have significantly degraded performance, the fingerprint similarity still remains high, demonstrating the robustness of our scheme against such fine-tuning attacks.

### 5.2.2 ROBUSTNESS AGAINST PRUNING

An attacker may attempt to disrupt the fingerprint by increasing pruning levels. Therefore, we examine the impact of varying pruning ratios on fingerprint robustness. Specifically, we prune Llama2-7B with SliceGPT (Ashkboos et al. (2024)), a structured pruning by removing entire rows or columns from LLM weight matrices while minimizing performance degradation, and analyze fingerprint similarity and performance variations across different pruning levels.

Figure 3c shows that the PPL of Llama2-7B on PTB dataset deteriorates beyond that of an LSTM baseline when the pruning ratio reaches 0.25, but our method maintains a high fingerprint similarity ($> 0.9$) for infringement detection. We provide more detection results for pruned model instances in Appendix C.

## 5.3 COMPARISON

This section benchmarks SELF against published structural fingerprinting methods for LLMs, including PCS and ICS proposed by Zeng et al. (2024), and REEF proposed by Zhang et al. (2025). A detailed introduction of these related works are presented in Appendix A.

Table 2 compares the mechanism and fingerprint size against various modifications across these LLM fingerprinting methods. SELF employs SVD- and EVD- based invariant matrix decomposition, requiring only partial parameters for fingerprint extraction and achieving a compact fingerprint size of approximately $10^3$ elements (for any model size). Compared to prior works that extract fingerprints ranging from $10^4$ to $10^9$ elements (for a 7B model), SELF significantly reduces storage overhead while preserving discriminative capability.

Table 2: Comparison of Mechanism and Fingerprint Size

| Method | Requirement | Fingerprint size |
|---|---|---|
| PCS | Parameter's vector direction | parameter_num ($\sim 10^9$) |
| ICS | Invariant terms' similarity | Partial parameters and input | $3 \times$ selected_layer_num $\times$ (sample_num)$^2$ ($\sim 10^9$) |
| REEF | Representation's CKA | Partial parameters and input | sample_num $\times d_{\text{model}}$ ($\sim 10^4 - 10^6$) |
| SELF | Invariant matrix decomposition | Partial parameters | $4N_{\mathcal{F}} \times h$ ($\sim 10^3$) |

CKA: Centered Kernel Alignment

Table 3 compares robustness against various attacks. By leveraging intrinsic model weights and eliminating input dependency, SELF is inherently robust against false claim attacks. Its SVD- and EVD- constructed fingerprints also provide strong attack resilience against linear mapping and permutation attacks due to their fundamental mathematical invariance properties. Besides, the proposed method is also evaluated to be robust against fine-tuning and pruning attacks, highlighting its effectiveness in practical deployment scenarios. A quantitative comparison between SELF and these methods are further provided in Appendix C.

Table 3: Comparison of Robustness Against Various Attacks

| Methods | Fine-tuning | Pruning | Permutation Attack | Linear-mapping Attack | False Claim Attack |
|---|---|---|---|---|---|
| PCS | ✓ | | | | ✓ |
| ICS | ✓ | | ✓ | ✓ | |
| REEF | ✓ | ✓ | ✓ | ✓ | |
| SELF | ✓ | ✓ | ✓ | ✓ | ✓ |

✓ indicates resilience against this attack.

## 6 CONCLUSION

This paper presents SELF, a weight-based fingerprinting method that eliminates input dependency, preventing false claim attacks. By extracting fingerprints utilizing singular values and eigenvalues, we leverage their mathematical properties to guarantee uniqueness, scalability, and robustness. A *SimNet* further learns distinctive fingerprint patterns, enabling effective IP infringement detection. Our approach thus serves as a reliable tool for LLM model IP protection.

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

APPENDIX

## A  RELATED WORKS

Current model IP infringement detection methods mainly use watermarking or fingerprinting. Watermarking embeds identifiable features into the target models, while fingerprinting extracts unique identifiers without modifying the model, thus avoiding retraining and performance degradation.

**Watermarking**  Watermarking approaches for LLMs primarily operate through two paradigms: model-centric watermarking and sampling-centric watermarking.

Model-centric techniques focus on embedding watermarks by modifying the model's architecture, parameters, or training process, typically involve fine-tuning or quantization strategies. For instance, Xu et al. (2024a) leverages the permutation equivariance property of Transformers to embed watermarks by fine-tuning on strategically permuted data samples. Alternatively, Li et al. (2024) introduces a knowledge injection approach where the watermark is carried through learned knowledge, while Li et al. (2023) implements watermarking during the model quantization process. These approaches typically require additional retraining or fine-tuning cost.

Sampling-centric methods embed watermarks by modifying the token sampling process or latent representations during text generation. These approaches do not alter the model's parameters but instead tweak the decoding strategy or latent space. Zhang et al. (2024) presents REMARK-LLM, a framework that encodes both generated text and binary messages into latent space before transforming tokens into sparse distributions. Dathathri et al. (2024) develops Tournament sampling, a modified sampling algorithm for watermark embedding. The KGW method (Kirchenbauer et al. (2023)) employs a "green list" approach, preferentially selecting predetermined tokens during generation. However, such watermarking scheme remains susceptible to editing and spoofing attacks Sadasivan et al. (2025). To mitigate this issue, Liu et al. (2024a) proposes a semantic invariant watermarking (SIR) method that generates watermarked tokens based on sentence-level embeddings. For sampling-centric approaches, additional post-processing may be required to verify watermarks.

**Fingerprinting**  Existing LLM fingerprinting methods encompass either behavior fingerprinting and structure fingerprinting. Behavior fingerprinting focus on identifying LLMs by analyzing their output characteristics or input-response patterns. These methods leverage the unique behavior signatures that emerge from how models respond to specific inputs or adversarial prompts. Gubri et al. (2024) employs adversarial suffixes to elicit model-specific responses, while Iourovitski et al. (2024) develops an evolutionary strategy using one LLM to identify distinctive features of others. Pasquini et al. (2025) utilizes carefully designed queries to detect specific model versions. As these methods rely on observable input-output interactions, they are effective for black-box scenarios but remain vulnerable to attacks like false claim attack Liu et al. (2024b).

Structural fingerprinting methods analyze internal parameters or activation patterns. These methods exploit the unique structural properties of a model's architecture or training dynamics. Zeng et al. (2024) proposes HuRef, a human-readable fingerprint based on the stability of LLM parameter vector directions after pretraining convergence. While these vector directions demonstrate resilience to subsequent training, they remain vulnerable to direct weight manipulations such as permutation and linear mapping. To address this limitation, the authors develop three rearrangement-invariant fingerprint terms. Zhang et al. (2025) introduces REEF, a representation-based fingerprinting method employing Centered Kernel Alignment (CKA) to compute similarity score. Wu et al. (2025) presents TensorGuard, a framework extracting gradient-based signatures by analyzing their internal gradient responses across tensor layers to random input perturbations. However, these fingerprinting techniques also exhibit vulnerability to false claim attacks Liu et al. (2024b) due to their inherent reliance on input engagement.

## B IMPLEMENTATION OF FALSE CLAIM ATTACK ON HUREF

In this section, we simulate an adversarial scenario by conducting false claim attacks against the HuRef (Zeng et al. (2024))'s ICS method. Figure 4 shows the critical path in the similarity scoring mechanism of HuRef. Given an LLM, HuRef uses the input embeddings $E$ together with the model weights to compute invariant terms $I$, which serve as the model's fingerprint. The cosine similarity of $I$ is then used to detect potential IP infringement. In addition, HuRef encodes $I$ into feature vector $v$ and generate human-readable fingerprints (images), where visual similarity indicates related models.

We envisioned an attack scenario where the malicious actor (a malicious owner or model stealer) aims to falsely claim the ownership of an unrelated model to his/her target or stolen model. To this end, the attacker carefully crafts input tokens so that, when the feature vector is computed following HuRef, the unrelated model yield high similarity to attacker's model, leading to deceptive readable fingerprints. Since the embedding step ($X \rightarrow E$) is not differentiable, we choose the genetic algorithm to implement the attack. Specifically, we treat a token as a gene and a input as an individual, and define the fitness as the similarity between the feature vectors computed by the two models given that input. In each generation, we apply random gene mutations (i.e., token substitutions) to the population of input individuals and select those with the highest fitness (i.e., highest feature vector similarity) for crossover to produce a new population. After several generations, the input with the highest fitness in the final population is selected as the false claim input. Such an attack only manipulates the input without altering the model itself.

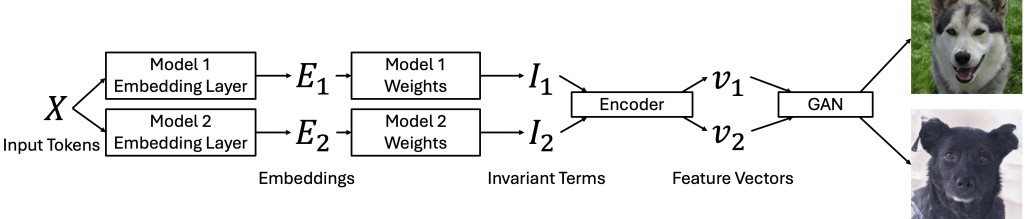

Figure 4: HuRef calculation process in our false claim attack

In our experiment, we assume the attacker holds Llama2-7B and he/she wants to deliberately claim the ownership of an unrelated model Qwen1.5-7B. By feeding the two models with a carefully crafted input obtained as described above, their feature vector similarity increased from 28% to 95% after 100 generations, leading to a false claim of model piracy. Moreover, the generated readable fingerprints shifted from being completely unrelated to appearing similar when using the false claim tokens, indicating that fingerprint similarity can be artificially induced.

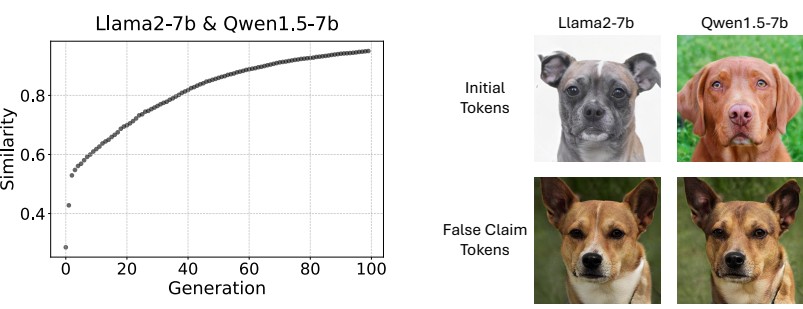

Feature vectors' similarity change     Human-readable fingerprint change

Figure 5: False-claim attack on HuRef: both the feature vector and the human-readable fingerprint of the unrelated model (Qwen1.5-7B) become highly similar to that of the target model (Llama2-7B).

# C    QUANTITATIVE COMPARISON WITH PREVIOUS METHODS

To further demonstrate the numerical superiority of our method, this section presents quantitative comparisons between our method with previous approaches. Follow Zhang et al. (2025), we present the IP detection results on publicly available models in Table 4 to Table 7. The performance (similarity) for previous methods are sourced directly from Zhang et al. (2025), which conducted comprehensive comparison with existing works and reported the corresponding data. For consistency, we reuse these reported values to benchmark our method against the state of the art. False judgments are highlighted in gray.

**Comparison of fine-tuned variants detectability**    Table 4 compares SELF with existing methods for detecting fine-tuned related models. The target model is Llama2-7B and its six fine-tuned variants are set as suspect models, fine-tuned on datasets ranging from 5M to 700B tokens. It can be observed that even after the target model is fine-tuned on large-scale data, SELF still identifies related models with a similarity score above 0.9, while PCS and ICS failed to detect some fine-tuned related models.

Table 4: Comparison of fingerprint similarity score on Llama2-7B fine-tuned models

| | Fine-tuning | | | | | |
| --- | --- | --- | --- | --- | --- | --- |
| | Llama-2-finance-7B (5M Tokens) | Vicuna-1.5-7B (370M Tokens) | Wizardmath-7B (1.8B Tokens) | Chinesellama-2-7B (13B Tokens) | Codellama-7B (500B Tokens) | Llemma-7B (700B Tokens) |
| PCS | 0.9979 | 0.9985 | 0.0250 | 0.0127 | 0.0105 | 0.0098 |
| ICS | 0.9952 | 0.9949 | 0.9994 | 0.4996 | 0.2550 | 0.2257 |
| REEF | 0.9950 | 0.9985 | 0.9979 | 0.9974 | 0.9947 | 0.9962 |
| SELF | 0.9948 | 0.9949 | 0.9949 | 0.9947 | 0.9641 | 0.9699 |

**Comparison of pruned variants detectability**    Tables 5 and 6 compares SELF with existing methods for detecting pruned related models. Llama-2-7B is used as the target model and its pruned variants as suspect models. Table 5 lists the fingerprint similarity scores of Llama-2-7B's structured pruned models and Table 6 reports those of unstructured pruned models. The results show that SELF can reliably detect pruned related models with high similarity.

Table 5: Comparison of fingerprint similarity score on Llama2-7B structured pruned models

| | Structured Pruning | | | | | |
| --- | --- | --- | --- | --- | --- | --- |
| | Sheared-Llama-1.3B-pruned | Sheared-Llama-1.3B | Sheared-Llama-1.3B-sharegpt | Sheared-Llama-2.7B-pruned | Sheared-Llama-2.7B | Sheared-Llama-2.7B-sharegpt |
| PCS | 0.0000 | 0.0000 | 0.0000 | 0.0000 | 0.0000 | 0.0000 |
| ICS | 0.4927 | 0.3512 | 0.3510 | 0.6055 | 0.4580 | 0.4548 |
| REEF | 0.9368 | 0.9676 | 0.9710 | 0.9278 | 0.9701 | 0.9991 |
| SELF | 0.9810 | 0.9896 | 0.9897 | 0.9853 | 0.9914 | 0.9948 |

*-pruned*: the version without continued pre-training;
*-sharegpt*: the version with instruction tuning.

Table 6: Comparison of fingerprint similarity score on Llama2-7B unstructured pruned models

| | Unstructured Pruning | | |
| --- | --- | --- | --- |
| | Sparse-Llama-2-7B | Wanda-Llama-2-7B | GBLM-Llama-2-7B |
| PCS | 0.9560 | 0.9620 | 0.9616 |
| ICS | 0.9468 | 0.9468 | 0.9478 |
| REEF | 0.9985 | 0.9986 | 0.9991 |
| SELF | 0.9950 | 0.9953 | 0.9954 |

**Comparison of merged models detectability**    In the model merging scenario, a suspect model is often derived from multiple victim models via different techniques. Table 7 presents the fingerprint similarity scores between weight-merged model (EvoLLM-jp-7B), distribution-merged model (FuseLLM-7B), and their corresponding victim target models. The results demonstrate that SELF reliably detects weight merged models as related. For distribution merging case, however, SELF

successfully identifies the relatedness between FuseLLM-7B and its base model (Llama-2-7B), but fails to link it to OpenLlama2-7B and MPT-7B. We analyze this limitation in the following paragraphs and highlight REEF's weakness, despite its higher similarity score in Table 7.

Table 7: Comparison of fingerprint similarity score on merged models

| Target models | Weight Merging (EvoLLM-jp-7B) | | | Distribution Merging (FuseLLM-7B) | | |
|---|---|---|---|---|---|---|
| | Shisa-gamma-7B-v1 | Wizard-math-7B-1.1 | Abel-7B-002 | Llama-2-7B | OpenLlama2-7B | MPT-7B |
| PCS | 0.9992 | 0.9990 | 0.9989 | 0.9997 | 0.0194 | 0.0000 |
| ICS | 0.9992 | 0.9988 | 0.9988 | 0.1043 | 0.2478 | 0.1014 |
| REEF | 0.9635 | 0.9526 | 0.9374 | 0.9996 | $0.6713^*$ | $0.6200^*$ |
| SELF | 0.9693 | 0.9911 | 0.9962 | 0.9949 | 0.0174 | 0.0051 |

$^*$ indicates a potential judgment as unrelated.

**High Risk of False Positives in REEF**  Although REEF achieves similarity scores above 0.62 for the related models (OpenLlama2-7B and MPT-7B) in Table 7, it exhibits a high risk of misclassifying unrelated models as related. To demonstrate this, we construct a focused comparison using Llama2-7B and seven distinct unrelated models (Table 8). Following the settings in Zhang et al. (2025), we evaluate REEF using representation-based fingerprinting (CKA on the $18_{th}$ layer) across 200 samples from five datasets: TruthfulQA (Lin et al. (2022)), ConfAIde (Mireshghallah et al. (2024)), PKU-SafeRLHF (Ji et al. (2025)), ToxiGen (Hartvigsen et al. (2022)), and SST2 (Socher et al. (2013)).

As shown in Table 8, REEF's similarity scores for unrelated models are inconsistently high, exceeding **0.67** on the SST2 dataset. This suggests that a threshold set to detect OpenLlama2-7B (0.6713) would incorrectly flag unrelated models as infringed. In contrast, our SELF method maintains consistently low scores ($<0.3$) for unrelated models, demonstrating superior robustness against false claim attacks.

Table 8: Comparison of Llama2-7B unrelated models fingerprint similarity output

| Method | Dataset | Mistral-7B | MPT-7B | InternLM2.5 | GPT2-Large | Cerebras-GPT | Pythia-6.7B | OPT-6.7B |
|---|---|---|---|---|---|---|---|---|
| REEF | TruthfulQA | 0.1975 | 0.2111 | 0.1942 | 0.2320 | 0.2257 | 0.2307 | 0.2437 |
| | ConfAIde | 0.2340 | 0.2449 | 0.2308 | 0.2464 | 0.2471 | 0.2723 | 0.2419 |
| | PKU-SafeRLHF | 0.5099 | 0.5150 | 0.4307 | 0.4808 | 0.4390 | 0.4844 | 0.4764 |
| | ToxiGen | 0.5894 | 0.5528 | 0.6067 | 0.5366 | 0.6358 | 0.6151 | 0.4716 |
| | SST2 | 0.6772 | 0.6221 | 0.6628 | 0.6034 | 0.5945 | 0.6783 | 0.5982 |
| SELF | | 0.0050 | 0.0050 | 0.0050 | 0.2372 | 0.2874 | 0.0002 | 0.0009 |

Furthermore, REEF's performance on *true* victim models is also sensitive to the dataset used. Table 9 shows the similarity output of REEF when verifying the actual source models of FuseLLM-7B on the problematic datasets (PKU-SafeRLFH, ToxiGen and SST2) identified above. For example, on ToxiGen, the score for the true victim model OpenLlama2-7B drops to **0.5614**, which is indistinguishable from the scores of unrelated models (e.g., Mistral-7B at 0.5894 in Table 8). This overlap confirms that REEF lacks a clear decision boundary to distinguish distribution-merged models from unrelated ones, whereas SELF demonstrates consistent discriminative capability across all test scenarios.

Table 9: Similarity output of REEF on FuseLLM-7B's source models across different datasets

| Model | PKU-SafeRLHF | ToxiGen | SST2 |
|---|---|---|---|
| OpenLlama2-7B | 0.5265 | 0.5614 | 0.6356 |
| MPT-7B | 0.5140 | 0.6360 | 0.7317 |

**Our hypothesize**  We hypothesize that the challenges faced by all methods in detecting FuseLLM-7B's relation to OpenLlama2-7B and MPT-7B stems from the unique fusion mechanism of distribution merging. The distribution merged model does not directly steal its victim models' weights; instead, it is trained using the output distribution of the victim models. In the case of FuseLLM-7B, Llama-2-7B is adopted as the base model while OpenLlama2-7B and MPT-7B only provide

output distribution for training. Consequently, methods based on model weight analysis may still detect merged model's relatedness to its base model but are extremely difficult to detect such output distributed-based infringement. To address such limitation, we propose that future detection frameworks adopt a hybrid approach, combining weight-based metrics for base model identification with output-consistency checks for distilled knowledge. This dual strategy would ensure robust detection coverage across both architectural reuse and distribution-based fusion scenarios.

## D  EXPLORING FINGERPRINT DISTANCE FOR DISCRIMINATION

This section evaluates SELF's effectiveness across routine models using fingerprint distance. Routine models are defined as publicly available models with known relationships (related or unrelated) to the target model, as listed in Table 1 of Section 5.1. The L2 distance between the fingerprints extracted from the target model and its related models (fine-tuned, pruned, quantized, or distilled variants) and unrelated models is computed. A smaller fingerprint distance indicates a higher likelihood that the two models are related.

Table 10 reports the distance between fingerprints extracted from the target models (Qwen2.5-7B and Llama2-7B) and their respective suspect models (both related and unrelated). For both target models, their distance exhibit a clear separation between related and unrelated cases (>0.95 for unrelated models and <0.65 for related models), with a sufficient margin (0.3) validating the discriminative power of our fingerprinting method. While our approach reliably detects IP infringement using fingerprint distance, the margin between the two categories is notably smaller than the margin (>0.7) achieve with *SimNet*, further underscoring the importance and necessity of *SimNet*. Additional evaluation and discussion can be found in Section E.5.

Table 10: Fingerprint Distance (Target Models: Qwen2.5-7B and Llama2-7B)

| **Qwen2.5-7B Unrelated** | | **Qwen2.5-7B Related** | | **Llama2-7B Unrelated** | | **Llama2-7B Related** | |
|---|---|---|---|---|---|---|---|
| **Model** | **Distance** | **Model** | **Distance** | **Model** | **Distance** | **Model** | **Distance** |
| Mistral-7B-V0.3 | 1.3050 | **Fine-tuned Variants** | | Mistral-7B-V0.3 | 0.9675 | **Fine-tuned Variants** | |
| Llama2-7B | 1.5112 | Qwen2.5-7B-Instruct | 0.0298 | Qwen1.5-7B | 1.2017 | Llama-2-7B-Chat | 0.0408 |
| Baichuan2-7B | 2.0768 | Qwen2.5-Math-7B | 0.4692 | Baichuan2-7B | 1.5762 | CodeQwen2.5-7B | 0.6358 |
| InternLM2.5-7B | 1.5717 | Qwen2.5-Coder-7B | 0.4946 | InternLM2.5-7B | 1.6968 | Llemma-7B | 0.5739 |
| GPT2-Large | 1.6143 | TableGPT2-7B | 0.0528 | GPT2-Large | 1.4820 | **Pruned Variants** | |
| Cerebras-GPT-1.3B | 2.0006 | Qwen2.5-7B-Medicine | 0.0328 | Cerebras-GPT-1.3B | 1.6188 | Sheared-Llama-2.7B | 0.4376 |
| ChatGLM2-6B | 1.9483 | Qwen2.5-7B-abliterated-v2 | 0.0307 | ChatGLM2-6B | 1.6980 | SparseLlama-2-7B | 0.1961 |
| OPT-6.7B | 2.9490 | **Quantized Variants** | | OPT-6.7B | 2.6365 | **Quantized Variants** | |
| Pythia-6.9B | 1.5799 | Qwen2.5-7B-4bit | 0.0002 | Pythia-6.9B | 2.3580 | Llama2-7B-4bit | 0.0005 |
| MPT-7B | 2.1532 | Qwen2.5-7B-8bit | 0.0002 | MPT-7B | 1.6858 | Llama2-7B-8bit | 0.0005 |

Table 11 reports the pairwise fingerprint distances between the ten unrelated models listed in Table 10. It can be observed that the fingerprint distance among these unrelated models are all >1. This confirms that our method can accurately distinguish unrelated models and thus avoid false positives.

Table 11: Fingerprint Distance between Unrelated Models

| | Mis | Qwe | Bai | Int | Gpt | Cer | Cha | OPT | Pyt | MPT |
|---|---|---|---|---|---|---|---|---|---|---|
| Mistral-7B-V0.3 | | 1.0114 | 1.8145 | 1.5130 | 1.7239 | 1.8124 | 1.5797 | 2.8946 | 2.2746 | 1.9419 |
| Qwen1.5-7B | 1.0114 | | 1.5932 | 1.5750 | 1.5081 | 1.8140 | 1.9056 | 2.6926 | 1.9453 | 1.8204 |
| Baichuan2-7B | 1.8145 | 1.5932 | | 2.5250 | 1.8923 | 1.9126 | 2.2444 | 2.5207 | 2.8323 | 1.8068 |
| InternLM2.5-7B | 1.5130 | 1.5750 | 2.5250 | | 2.4088 | 2.6447 | 2.4530 | 3.5071 | 1.5415 | 2.4680 |
| GPT2-Large | 1.7239 | 1.5081 | 1.8923 | 2.4088 | | 1.2487 | 1.8379 | 2.0373 | 2.6276 | 1.7582 |
| Cerebras-GPT-1.3B | 1.8124 | 1.8140 | 1.9126 | 2.6447 | 1.2487 | | 1.8842 | 1.9259 | 2.9951 | 1.9990 |
| ChatGLM2-6B | 1.5797 | 1.9056 | 2.2444 | 2.4530 | 1.8379 | 1.8842 | | 2.7989 | 3.1283 | 2.1770 |
| OPT-6.7B | 2.8946 | 2.6926 | 2.5207 | 3.5071 | 2.0373 | 1.9259 | 2.7989 | | 3.7288 | 2.4523 |
| Pythia-6.9B | 2.2746 | 1.9453 | 2.8323 | 1.5415 | 2.6276 | 2.9951 | 3.1283 | 3.7288 | | 2.8906 |
| MPT-7B | 1.9419 | 1.8204 | 1.8068 | 2.4680 | 1.7582 | 1.9990 | 2.1770 | 2.4523 | 2.8906 | |

To validate the effectiveness of our method on distilled models, we evaluated three DeepSeek distillation models of varying sizes and architectures. As shown in Table 12, their distance from respective base models remain < 0.3, confirming their relatedness.

Table 12: Fingerprint Distance of DeepSeek models

| Base Model | Distilled Model | Distance |
|---|---|---|
| Qwen2.5-Math-1.5B | DeepSeek-R1-Distill-Qwen-1.5B | 0.1756 |
| Qwen2.5-Math-7B | DeepSeek-R1-Distill-Qwen-7B | 0.1266 |
| Llama-3.1-8B | DeepSeek-R1-Distill-Llama-8B | 0.2748 |

# E  DETAILS OF *SimNet*

This section presents the implementation details of *SimNet*, including its network architecture and training settings, as well as its impact on SELF's robustness. The input of the SimNet is the suspect model's fingerprint $F_S$, and the output is the similarity score range in [0, 1].

## E.1  ARCHITECTURE

The overall architecture of the *SimNet* is shown in Table 13.

Table 13: Neural Network Architecture of SimilarityNet (Input Shape: $(B, 4N_{\mathcal{F}}, h)$)

| Component | Type | Input Shape | Output Shape |
|---|---|---|---|
| Input | - | $(B, 4N_{\mathcal{F}}, h)$ | - |
| unsqueeze | Dimension Expansion | $(B, 4N_{\mathcal{F}}, h)$ | $(B, 1, 4N_{\mathcal{F}}, h)$ |
| conv1 + bn1 + ReLU | Conv2d, BatchNorm2d, Activation | $(B, 1, 4N_{\mathcal{F}}, h)$ | $(B, 64, 4N_{\mathcal{F}}, h)$ |
| layer1 | ResidualBlock ×2, Stride 1 | $(B, 64, 4N_{\mathcal{F}}, h)$ | $(B, 64, 4N_{\mathcal{F}}, h)$ |
| layer2 | ResidualBlock ×2, Stride 2 | $(B, 64, 4N_{\mathcal{F}}, h)$ | $(B, 128, 2N_{\mathcal{F}}, h/2)$ |
| layer3 | ResidualBlock ×2, Stride 2 | $(B, 128, 2N_{\mathcal{F}}, h/2)$ | $(B, 256, N_{\mathcal{F}}, h/4)$ |
| layer4 | ResidualBlock ×2, Stride 2 | $(B, 256, N_{\mathcal{F}}, h/4)$ | $(B, 512, N_{\mathcal{F}}/2, h/8)$ |
| layer5 | ResidualBlock ×2, Stride 1 | $(B, 512, N_{\mathcal{F}}/2, h/8)$ | $(B, 512, N_{\mathcal{F}}/2, h/8)$ |
| avgpool | AdaptiveAvgPool2d (1, 1) | $(B, 512, N_{\mathcal{F}}/2, h/8)$ | $(B, 512, 1, 1)$ |
| flatten (view) | View | $(B, 512, 1, 1)$ | $(B, 512)$ |
| fc | Linear | $(B, 512)$ | $(B, 1)$ |
| sigmoid | Activation | $(B, 1)$ | $(B, 1)$ |
| squeeze | Dimension Squeeze | $(B, 1)$ | $(B, )$ |

In our experiments, $N_{\mathcal{F}}$=8, $h$=256.

## E.2  DATASETS

The *SimNet* training datasets for each target model includes the following models' fingerprints:

For Qwen2.5-7B: Qwen2.5-7B and its augmented model (label 1), Qwen2.5-7B-Instruct and its augmented model (label 1), MPT-7B and its augmented model (label 0), GPT2-Large and its augmented model (label 0), and Mistral-7B-v0.3 and its augmented model (label 0).

For Llama2-7B: Llama2-7B and its augmented model (label 1), Llama2-7B-chat (label 1) and its augmented model, MPT-7B and its augmented model (label 0), InternLM2.5-7B and its augmented model (label 0), and Mistral-7B-v0.3 and its augmented model (label 0).

For Shisa-gamma-7B-v1: Shisa-gamma-7B-v1 and its augmented model (label 1), MPT-7B and its augmented model (label 0), InternLM2.5-7B and its augmented model (label 0), Llama2-7B and its augmented model (label 0).

For Wizard-math-7B-1.1: Wizard-math-7B-1.1 and its augmented model (label 1), MPT-7B and its augmented model (label 0), InternLM2.5-7B and its augmented model (label 0), Llama2-7B and its augmented model (label 0).

For Abel-7B-002: Abel-7B-002 and its augmented model (label 1), MPT-7B and its augmented model (label 0), InternLM2.5-7B and its augmented model (label 0), Llama2-7B and its augmented model (label 0).

For OpenLlama2-7B: OpenLlama2-7B and its augmented model (label 1), MPT-7B and its augmented model (label 0), InternLM2.5-7B and its augmented model (label 0), Llama2-7B and its augmented model (label 0).

For MPT-7B: MPT-7B and its augmented model (label 1), Mistral-7B-v0.3 and its augmented model (label 0), InternLM2.5-7B and its augmented model (label 0), Llama2-7B and its augmented model (label 0).

### E.3 HYPERPARAMETER

The model was trained using the AdamW optimizer with an initial learning rate of $1 \times 10^{-4}$ and weight decay of $1 \times 10^{-6}$, combined with a step learning rate scheduler (step size = 100, $\gamma = 0.8$). We minimized the binary cross-entropy loss with label smoothing ($\eta = 0.01$) and incorporated adversarial training through gradient sign perturbations ($\epsilon = 1 \times 10^{-5}$). Training proceeded for 1000 epochs, using PyTorch's default parameter initialization scheme and random seed 42.

For each augmentation type, we generate three augmented models with the following parameters:

- Gaussian Noise: The strength $\alpha$ is 0.1, 1, 10.
- Row Deletion: The number of deleted rows $n_r$ is 10, 100, 1000.
- Column Deletion: The number of deleted columns $n_r$ is 10, 100, 1000.
- Random Masking: The mask rate $r$ is 0.1, 0.25, 0.5.

### E.4 OVERHEAD

*SimNet* requires only one-time training per target model and can be reused for all subsequent detections, ensuring that the computational overhead is amortized over time. Training *SimNet* takes about 2.5 minutes using a single RTX4090 GPU, while one inference takes less than 0.1 seconds. Despite this minimal overhead, our experiments demonstrated that *SimNet* delivers superior performance compared to existing methods, striking an optimal cost-benefit balance for practical deployment.

### E.5 IMPACT OF *SimNet* ON ROBUSTNESS

This section evaluates the impact of employing *SimNet* on method robustness. Using Llama2-7B as the target model, we evaluate both the distance and the similarity score in detecting related models obtained via the commonly used fine-tuning and pruning attacks.

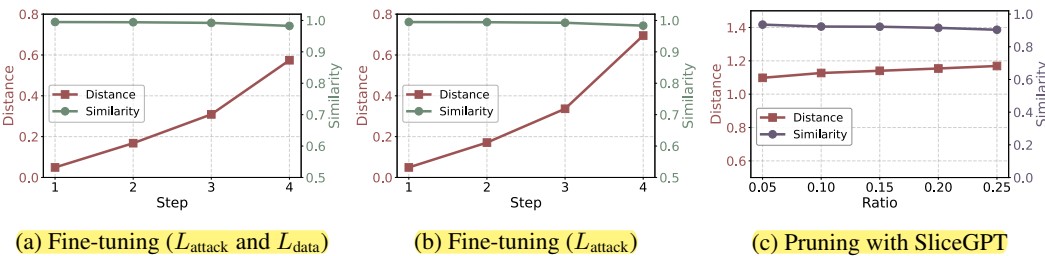

(a) Fine-tuning ($L_{\text{attack}}$ and $L_{\text{data}}$)   (b) Fine-tuning ($L_{\text{attack}}$)   (c) Pruning with SliceGPT

Figure 6: Fingerprint similarity and distance change of Llama2-7B under different attacks. (a) and (b) show the results under fine-tuning attacks. (c) shows the results under SliceGPT pruning.

**SimNet enhances SELF's robustness**   As shown in Figure 6a and 6b, as fine-tuning steps increase, the distance (ideal: small for related models) between Llama2-7B (target model) and its variant grows quickly, while the similarity (ideal: high for related models) score remains consistently stable high – highlighting *SimNet*'s superior robustness in detecting fine-tuned models. For pruning attacks (Figure 6c), *SimNet* maintains high similarity scores ($>0.9$) even when the distance falsely indicted unrelated models ($>1$). This further confirms that *SimNet* can capture intrinsic features beyond distance, thereby ensuring the robustness of SELF against adversarial modifications.

# F   ABLATION STUDY ON FINGERPRINT EXTRACTION

In this section, we assess how different settings affect fingerprint extraction. We define the **fingerprint margin** as the difference between: 1) the minimum L2 distance among unrelated models' fingerprints, and 2) the maximum L2 distance among related models' fingerprints. Here, we have considered all models appeared in Tables 10 to 12. A large positive margin indicates clearer distinguishability between related and unrelated models, while a negative margin suggests that some unrelated models exhibit smaller fingerprint distance than the maximum observed among related models. Accordingly, we systematically investigate the impact of the following settings on the fingerprint margin:

1. Invariant matrix construction choice (singular values or eigenvalues)
2. Transformer block layers selection (e.g. early vs. late layers)
3. Weight selection (e.g. $W_Q$ and $W_K$ vs. $W_V$ and $W_O$)
4. Parameter $h$ (top-$h$ singular values / eigenvalues are selected to form the fingerprint)
5. Parameter $N_\mathcal{F}$ (how many layers are selected to form the fingerprint)

As shown in Table 14, the first eight layers offer larger fingerprint margins, indicating stronger differentiation between related and unrelated models than other layers. Although using only $W_Q$ and $W_K$ can achieve nearly comparable margin, to ensure that the fingerprint contains more information and thereby guarantees the performance of *SimNet*, we also include $W_V$ and $W_O$ as a source for fingerprint extraction.

Table 14: Fingerprint Margin in Different Weight Component Settings

| Invariant | Layer Selection | | | Weight Selection | | |
|---|---|---|---|---|---|---|
| | First 8 Layers | Middle 8 Layers | Last 8 Layers | $W_Q, W_K$ | $W_V, W_O$ | Both |
| Singular Value | 0.1484 | -0.1366 | -0.2020 | 0.1744 | -0.0744 | 0.1484 |
| Eigenvalue | 0.2362 | -0.0789 | -0.2475 | 0.2283 | -0.1538 | 0.2362 |
| Both | 0.3319 | -0.1380 | -0.1263 | 0.3382 | -0.0217 | 0.3319 |

Table 15 confirms that even small $h$ and $N_\mathcal{F}$ can provide discriminative capability, while larger $h$ and $N_\mathcal{F}$ leads to a greater fingerprint margin. The selection of $h$ should ensure robust fingerprint extraction while being practical for deployment across diverse model sizes, i.e., smaller than any model's $d$ and $d_{\text{model}}$. Thus, we chose $h = 256$ and $N_\mathcal{F} = 8$ in our experiments, which is compatible with most LLMs' weight dimensions.

Table 15: Fingerprint Margin under Different $h$ and $N_\mathcal{F}$ Settings

| Invariant | Varying $h$ | | | | Varying $N_\mathcal{F}$ | | | |
|---|---|---|---|---|---|---|---|---|
| | 32 | 64 | 128 | 256 | 2 | 4 | 6 | 8 |
| Singular Value | 0.0486 | 0.0731 | 0.1478 | 0.1484 | -0.0878 | 0.0816 | 0.1443 | 0.1484 |
| Eigenvalue | -0.0306 | 0.0339 | 0.1345 | 0.2362 | 0.0921 | 0.1375 | 0.1467 | 0.2363 |
| Both | 0.0653 | 0.1732 | 0.2624 | 0.3319 | 0.0911 | 0.2080 | 0.2794 | 0.3319 |

# G   LLM AND REPRODUCIBILITY STATEMENT

The authors acknowledge the use of LLMs for editorial assistance, specifically in grammar correction and language refinement. The LLM was not involved in the generation of ideas, design of experiments, and so on. All intellectual and scientific contributions presented in this manuscript are solely attributable to the authors.

We have released our code repository at https://anonymous.4open.science/r/SELF-BC5F, which contains the implementation details, fingerprint extraction results, and *SimNet* model parameters used in this work. Our method can be reproduced using the provided instructions and resources.

