# OpenReview forum: "SELF: A Robust Singular Value and Eigenvalue Approach for LLM  Fingerprinting"
_ICLR.cc/2026/Conference — Submitted to ICLR 2026_

### Official Review · Reviewer_HKUo · 2025-10-29

**Soundness:** 2
**Presentation:** 3
**Contribution:** 2
**Rating:** 2
**Confidence:** 4

**Summary:**

This paper presents a method for LLM intellectual property (IP) protection, aiming to overcome the vulnerabilities of current fingerprinting schemes, which are often susceptible to input-dependent false claim attacks or structural weight manipulations. The authors introduce SELF, a novel intrinsic weight-based fingerprinting scheme designed to be inherently resistant to false claim attacks by eliminating any dependency on model inputs. The method's core mechanism extracts transformation-invariant features by applying singular value (SVD) and eigenvalue (EVD) decomposition to specific matrices derived from the LLM's attention weights. This process yields a compact fingerprint that is theoretically robust to permutation and linear mapping attacks. To compare these fingerprints, the authors employ a neural network (SimNet), trained using a few-shot learning paradigm with data augmentation, to compute a final similarity score. Experimental results on Llama-7B, Llama2-7B, and their variants demonstrate that SELF maintains high detection accuracy and shows strong robustness against various modifications, including quantization, pruning, and fine-tuning attacks.

**Strengths:**

- This paper presents promising results.
- Perfect resistance against the False Claim Attack by eliminating dependency on inputs.
- This paper is generally easy to read.

**Weaknesses:**

- **On the Insufficient Justification for Robustness via Eq. 6 and 10:** The paper's theoretical motivation for robustness relies on Eq. 6 and Eq. 10. However, this analysis is incomplete. The authors argue that small perturbations $\Delta M$ lead to small changes in singular/eigenvalues. This overlooks the high dimensionality of LLM weight matrices (e.g., $a \times a$). Even if each element of $\Delta M$ is small ($k$), the spectral norm of the matrix ($||\Delta M||_2$) can have an upper bound as large as $ak$, which could be substantial.

- **On the Limited Applicability to Modern LLM Architectures:** The proposed invariant matrices in Eq. 12 and Eq. 14 are fundamentally dependent on the Multi-Head Attention (MHA) architecture, where $W_Q$ and $W_K$ have compatible dimensions. However, many modern and state-of-the-art LLMs (e.g., Llama 3, DeepSeek) have adopted more efficient attention mechanisms like Grouped-Query Attention (GQA) and Multi-Query Attention (MQA). In GQA/MQA, the key matrix has different dimensions from the query matrix, rendering Eq. 12 and 14 computationally invalid. This severely limits the method's applicability to many of the most relevant contemporary models.
- **On the Limited Scope of Experimental Validation:** The empirical evaluation is limited in scope. The method is only tested on two target models (Llama-7B and Llama2-7B) and primarily considers variants from fine-tuning and quantization. To make a convincing case for the method's practical robustness, it may be better for the authors to evaluate it against a wider range of common model modification techniques, such as knowledge distillation and model merging, which are known to significantly alter model weights.

- **On the Misleading Analysis of False Claim Attacks:** The paper's analysis of the "False Claim Attack" on HuRef (Appendix B) appears to be based on a non-standard definition. In fingerprinting literature, a False Claim Attack typically refers to the optimization of a transferable fingerprint that can be successfully claimed across many unrelated models. The attack demonstrated in the paper is a one-to-one attack, where an input is optimized to forge similarity between two specific models. This is more accurately described as an Ambiguity Attack, which is a known problem that can often be mitigated by methods like timestamping [1]. The paper provides no evidence that their attack on HuRef is transferable, thus weakening the claim that their input-independent method is necessary to solve this specific vulnerability.

- **On the Confusing and Unfavorable Baseline Comparison:** The comparison to baseline methods is presented in a confusing manner. The main body of the paper (Table 4) only compares SELF to REEF on unrelated models, while the full robustness comparison is relegated to Appendix F. More importantly, these appendix results suggest that SELF's performance is actually weaker than REEF's. This contradicts the paper's narrative of superior robustness and makes the justification for the new method unclear.

[1] False Claims against Model Ownership Resolution.

**Questions:**

Please refer to the Weaknesses section.

---

> ### Author Response · Authors · 2025-11-28
> **Response to Reviewer HKUo**
>
> **Comment 1**:
> On the Insufficient Justification for Robustness via Eq. 6 and 10: The paper's theoretical motivation for robustness relies on Eq. 6 and Eq. 10. However, this analysis is incomplete. The authors argue that small perturbations $\Delta M$ lead to small changes in singular/eigenvalues. This overlooks the high dimensionality of LLM weight matrices (e.g., $a \times a$). Even if each element of $\Delta M$ is small ($k$), the spectral norm of the matrix $(||\Delta M||_2)$ can have an upper bound as large as $ak$, which could be substantial.
>
> **Response**:
> We thank the reviewer for this insightful comment. While the theoretical upper bound of the spectral norm $||\Delta M||_2$ could scale with dimensionality, our method's robustness holds in practice for two key reasons:
>
> (1) Perturbations are practically bounded in related models, especially for earlier layer: Both Eq. 6 and Eq. 10 assume arbitrary small perturbations. In the case of creating functional related models, modifications (transfer learning, pruning, quantization, etc.) are constrained by performance preservation. As demonstrated in [ref1], earlier layers capture universal features that remain stable across transfer learning tasks. This implies that for the specific layers we utilize (the first 8 layers), the perturbation $\Delta M$ is structurally stable, ensuring a consistent positive fingerprint margin for earlier layers (demonstrated in Table 14 of our manuscript). For other modifications like pruning, many weights even remain unchanged.
>
> (2) Discriminability between related and unrelated models in high-dimensional space: Despite the increase in absolute distance caused by high dimensionality, the weight matrices of related and unrelated models remain distinguishable. For example, the L2 distances between attention weights $W_Q$ and $W_K$ (from first 8 layers) in Llama2-7B and its five related and unrelated models (3 related, 2 unrelated) demonstrate clear separation: As shown in the table below, related models exhibit obviously smaller perturbations in earlier layer attention weights, even for heavily fine-tuned related models (CodeLlama-7B and Llemma-7B). Our SimNet leverages this separability to achieve consistent discriminability, as validated by the experiments in our manuscripts.
>
> | Model Name | $W_Q$ | $W_K$ |
> |---|---|---|
> | Llama2-7B-chat (related) | 35.0354 | 34.7240 |
> | CodeLlama-7B (related) | 754.0130 | 777.6015 |
> | Llemma-7B (related) | 729.9012 | 755.2195 |
> | Qwen1.5-7B (unrelated) | 987.0753 | 1004.1936 |
> | Baichuan2-7B (unrelated) | 1237.8062 | 1255.2521 |
>
> [ref1] Howard, Jeremy, and Sebastian Ruder. *Universal Language Model Fine-tuning for Text Classification.* Proceedings of the 56th Annual Meeting of the Association for Computational Linguistics, 2018.

---

> ### Author Response · Authors · 2025-11-28
> **Response to Reviewer HKUo (2)**
>
> **Comment 2**:
> On the Limited Applicability to Modern LLM Architectures: The proposed invariant matrices in Eq. 12 and Eq. 14 are fundamentally dependent on the Multi-Head Attention (MHA) architecture, where $W_Q$ and $W_K$ have compatible dimensions. However, many modern and state-of-the-art LLMs (e.g., Llama 3, DeepSeek) have adopted more efficient attention mechanisms like Grouped-Query Attention (GQA) and Multi-Query Attention (MQA). In GQA/MQA, the key matrix has different dimensions from the query matrix, rendering Eq. 12 and 14 computationally invalid. This severely limits the method's applicability to many of the most relevant contemporary models.
>
> **Response**:
> Thank you for this comment. While our invariant matrices were initially derived for MHA, they are **fully compatible** with GQA and MQA through a simple transformation: by duplicating and expanding the key and value matrix in each group, ensuring that our computations remain applicable. This discussion has been incorporated into Section 5.1. To validate this, we have replaced Llama-7B with Qwen2.5-7B (GQA architecture) as the target model in Sec. 5.1. The results (summarized in the table below) confirm that our method achieves high distinguishability between related and unrelated Qwen2.5-7B variants, with similarity scores showing a clear separation ($>$0.9 for related models vs. $<$0.1 for unrelated models). This confirms the scalability of SELF across different architectures like GQA and MQA (where MQA is a special case of GQA with group number equals to 1), underscoring its broad applicability to modern LLMs.
>
> | **Model (Unrelated)** | **SimNet Score** | **Model (Related)** | **SimNet Score** |
> | :--- | :--- | :--- | :--- |
> | Mistral-7B-V0.3 | 0.0050 | **Fine-tuned Variants** | |
> | Llama2-7B | 0.0020 | Qwen2.5-7B-Instruct | 0.9950 |
> | Baichuan2-7B | 0.0009 | Qwen2.5-Math-7B | 0.9979 |
> | InternLM2.5-7B | 0.0139 | Qwen2.5-Coder-7B | 0.9910 |
> | GPT2-Large | 0.0050 | TableGPT2-7B | 0.9944 |
> | Cerebras-GPT-1.3B | 0.0011 | Qwen2.5-7B-Medicine | 0.9950 |
> | ChatGLM2-6B | 0.0091 | Qwen2.5-7B-abliterated-v2 | 0.9950 |
> | OPT-6.7B | 0.0005 | **Quantized Variants** | |
> | Pythia-6.9B | 0.0156 | Qwen2.5-7B-4bit | 0.9950 |
> | MPT-7B | 0.0050 | Qwen2.5-7B-8bit | 0.9950 |

---

> > ### Author Response · Authors · 2025-11-28
> > **Response to Reviewer HKUo (3)**
> >
> > **Comment 3**:
> > On the Limited Scope of Experimental Validation: The empirical evaluation is limited in scope. The method is only tested on two target models (Llama-7B and Llama2-7B) and primarily considers variants from fine-tuning and quantization. To make a convincing case for the method's practical robustness, it may be better for the authors to evaluate it against a wider range of common model modification techniques, such as knowledge distillation and model merging, which are known to significantly alter model weights.
> >
> > **Response**:
> > We appreciate the reviewer’s suggestion to expand the evaluation scope. To demonstrate broader applicability and practical robustness, (1) we added Qwen2.5-7B as a new target model family. Consistent IP detection results (Table 1) confirming that our method is effective across structurally different architectures; (2) we clarify that our method's robustness against model merging and knowledge distillation have already been evaluated, specifically:
> >
> > (2-a) Knowledge Distillation: We have also extended our evaluation to distilled models. In Table 12 of Appendix D, we evaluated the distance between the publicly available distilled DeepSeek models and their base models. The results demonstrate that our method remains effective in verifying the identity of models derived through such processes, provided the model architecture allows for weight mapping.
> >
> > (2-a) Weight-based Model Merging: We have explicitly evaluated robustness against model merging, a technique that interpolates weights between models. As detailed in Table 7 of Appendix C, our method successfully detects ownership in weight merged models.
> >
> > (2-c) Distribution-based Model Merging: Detecting distribution-based model merging is a fundamental challenge, as such merged models learn from output distributions rather than directly inherit weights. To the best of our knowledge, there is currently no weight-based fingerprinting method, including ours, capable of accurately identifying distribution merging models. Even REEF, the best-performance structural fingerprinting for comparison, shows ambiguous score overlaps (Table 8 and 9) between distribution merged and unrelated models, highlighting this as a challenge research problem. In the future work, we will explore hybrid fingerprints (e.g., combining weight-based fingerprints with activation-based watermarks to trace distillation) to address this gap. This discussion of limitation and future work has been appended to the last paragraph of Appendix C.

---

> > > ### Author Response · Authors · 2025-11-28
> > > **Response to Reviewer HKUo (4)**
> > >
> > > **Comment 4**:
> > > On the Misleading Analysis of False Claim Attacks: The paper's analysis of the "False Claim Attack" on HuRef (Appendix B) appears to be based on a non-standard definition. In fingerprinting literature, a False Claim Attack typically refers to the optimization of a transferable fingerprint that can be successfully claimed across many unrelated models. The attack demonstrated in the paper is a one-to-one attack, where an input is optimized to forge similarity between two specific models. This is more accurately described as an Ambiguity Attack, which is a known problem that can often be mitigated by methods like timestamping. The paper provides no evidence that their attack on HuRef is transferable, thus weakening the claim that their input-independent method is necessary to solve this specific vulnerability.
> > >
> > > **Response**: We appreciate the reviewer’s feedback regarding the taxonomy of attacks. To clarify, **our use of False Claim Attack (FCA) is appropriate and aligns with both the attack scope and empirical evidence**:
> > >
> > > (1) Distinction between Ambiguity attack and FCA: As studied in [ref2], ambiguity attack aims to cast doubt on the ownership verification by forging *additional* watermarks/fingerprints for an AI model in question, usually exploiting redundancy in watermarking/fingerprinting schemes. The goal of ambiguity attack is to create ownership uncertainty by enabling multiple parties to "prove" ownership. In contrast, FCA aims to falsely claim the ownership of an unrelated model by manipulating detection inputs. FCA exploits the vulnerability of input-dependent fingerprinting methods, which is precisely why an input-independent approach is necessary. **The two attacks are fundamentally different, as ambiguity attack creates multiple valid watermarks/fingerprints for one model, while FCA aims to create manipulated inputs to falsely link two or more unrelated models.** We claim that "transferrable adversarial inputs" are just one type of manipulated input. Though our HuRef example only forges similarity between two specific models, this attack aligns well with the attack objective (link two unrelated models) and method (manipulate input) of FCA, therefore, clearly belongs to the FCA category.
> > >
> > > [ref2] Fan, Lixin, et al. *Deepipr: Deep neural network ownership verification with passports.* IEEE Transactions on Pattern Analysis and Machine Intelligence, 2021.
> > >
> > > (2) Evidence of Transferability: we further claim that "one-to-many" FCA attacks can also be applied to input-dependent fingerprinting methods. As shown in Table 8 of Appendix C, a single set of adversarial inputs (SST2) can successfully trigger high similarity scores across 7 different, unrelated models under the REEF scheme. This confirms the possibility of a transferable attack—where a single adversarial input fools multiple unrelated models—thereby satisfying the definition of FCA originally proposed in [ref3] and underscoring the necessity of our proposed input-independent fingerprinting method.
> > >
> > > [ref3] Jian Liu, Rui Zhang, et al. *False claims against model ownership resolution*. 33rd USENIX Security Symposium, 2024.

---

> > > > ### Author Response · Authors · 2025-11-28
> > > > **Response to Reviewer HKUo (5)**
> > > >
> > > > **Comment 5**:
> > > > On the Confusing and Unfavorable Baseline Comparison: The comparison to baseline methods is presented in a confusing manner. The main body of the paper (Table 4) only compares SELF to REEF on unrelated models, while the full robustness comparison is relegated to Appendix F. More importantly, these appendix results suggest that SELF's performance is actually weaker than REEF's. This contradicts the paper's narrative of superior robustness and makes the justification for the new method unclear.
> > > >
> > > > **Response**:
> > > > We thank for the reviewer's feedback. We address the concern about baseline comparisons by reorganizing the paper’s structure and clarifying the interpretation of the robustness results.
> > > >
> > > > (1) Structural Reorganization: We have consolidated the comparison logic accordingly. Section 5.3 now presents a comprehensive mechanism-level comparison, while the quantitative comparison is moved to Appendix C due to page limitation. The mechanism-level qualitative comparison highlights that our method utilizes a significantly smaller fingerprint size and eliminates reliance on input data, thereby offering superior resistance to false claim attacks compared to REEF; The quantitative comparison further demonstrates our superior robustness against fine-tuning, pruning and weight merging attacks.
> > > >
> > > > (2) Clarification on "Weaker" Performance (Appendix C): Regarding the quantitative comparison on "Distribution merging" in Table 7 of Appendix C, we respectfully point out that higher raw similarity scores do not necessarily imply better robustness if distinctiveness is lost.
> > > >
> > > > The Problem with REEF: As detailed in Table 8 and Table 9, REEF exhibits a high false-positive risk. Specifically, REEF yields high similarity scores ($>$0.6) even for unrelated models on datasets like SST2. Consequently, REEF’s high score on "distribution merging" is ambiguous—it cannot reliably distinguish between a merged model (related model) and a completely unrelated one.
> > > >
> > > > Therefore, SELF and REEF exhibit comparable robustness against all evaluated attacks. However, SELF outperforms REEF in terms of smaller fingerprint size and resistance to false claim attacks, demonstrating SELF as a more practical and secure solution for model IP protection compared to existing methods.

---

### Official Review · Reviewer_QAmV · 2025-10-31

**Soundness:** 4
**Presentation:** 3
**Contribution:** 3
**Rating:** 6
**Confidence:** 4

**Summary:**

This paper proposes SELF, a robust fingerprinting method for large language models (LLMs) based on model weight singular values and eigenvalue decomposition, aiming to facilitate intellectual property protection. SELF extracts transformation-invariant fingerprint features from the attention layer weights of Transformer models and employs the few-shot learning-based SimNet to assess model similarity, thereby detecting model tampering and piracy. Experimental results demonstrate that SELF performs well under a variety of attack scenarios, such as weight perturbation, pruning, quantization, and fine-tuning, outperforming existing structure-based fingerprinting methods.

**Strengths:**

1. This work addresses the vulnerability of structure-based fingerprints against transformation attacks by introducing a purely weight-driven fingerprinting approach that does not rely on model inputs, fundamentally mitigating risks from adversarial model declaration attacks.
2. The method builds on the inherent invariance properties of singular values and eigenvalues to craft fingerprints robust to parameter perturbations, with clear mathematical underpinnings and theoretical justification.
3. The paper presents extensive experiments covering a wide range of practical adversarial scenarios in the LLM ecosystem—including pruning, quantization, merging, fine-tuning, and impersonation attacks—and conducts detailed comparisons with several baseline methods (PCS, ICS, REEF), providing comprehensive empirical validation.

**Weaknesses:**

1. Practical limitations: The method assumes a white-box setting where full access to all parameters of the target and suspected models is available. This assumption may not hold in real-world commercial or API-based protection scenarios where only black-box access is feasible.
2. It remains unclear whether SELF is robust to parameter perturbations in non-attention layers (e.g., MLP, LayerNorm). The authors could explore whether these components should be incorporated into fingerprint construction.
3. It is uncertain how the method adapts to structural variations in the model, such as changes in the number of layers (either adding or removing layers).

**Questions:**

1. Possible data leakage: Although the training and testing sets are nominally separated, the augmented samples used in training are highly similar to the original models, and the evaluation extensively involves these same or similar models and their fingerprints. This raises concerns about potential overlap in the feature distribution between training and test sets, possibly leading to overfitting. Furthermore, the generalization ability of SELF to unseen models has not been convincingly demonstrated. The authors are encouraged to clearly demarcate the training and testing model spaces and to include cross-family generalization experiments involving structurally diverse models to enhance credibility.
2. The method may be vulnerable to model distillation-based stealing attacks. In particular, results on FuseLLM-7B  in Appendix F suggest that SELF struggles to maintain robustness in such scenarios.

---

> ### Author Response · Authors · 2025-11-28
> **Response to Reviewer QAmV**
>
> We thank the reviewer for the valuable feedback. We have grouped the identified weakness and questions into the following comments for clarity and response.
>
> **Comment 1**:
> Practical limitations: The method assumes a white-box setting where full access to all parameters of the target and suspected models is available. This assumption may not hold in real-world commercial or API-based protection scenarios where only black-box access is feasible.
>
> **Response**:
> Thanks for the reviewer's comment. Our work focuses on IP forensic detection, where a trusted third party (TTA) can obtain temporary white-box access to both original and suspect models for model ownership investigation. This aligns with legal workflows, as exemplified by the real-world commercial case: Douyin (TikTok) v. B612 (2025) [ref1]. In this landmark case, the Beijing IP court (acting as the TTA) ruled in favor of Douyin after identifying structural and parametric similarities between Douyin's AI transformation model and B612's infringing model in its Kaji APP. Our method can provide accurate, robust, and efficient infringement detection for such scenarios.
>
> While white-box access restricts applicability to auditable or open-source models, our method offers critical advantages over black-box alternatives: (1) zero procedural overhead: our method does not require any retraining or model modifications, offering a lightweight and flexible solution which enables the protection of models that have been deployed in the markets; (2) enhanced security: our proposed fingerprints are constructed without relying on data input, eliminating the risk of forging fingerprint by crafting inputs, thus avoid false claim attacks.
>
> Accordingly, we have revised our threat model in Sec. 4.1 to clearly explain the TTA's role and forensic workflow of our proposed work.
>
> [ref1]: China's first case on protecting structure and parameters of AI models: Douyin v. B612 Kaji. <https://www.linkedin.com/pulse/china-first-case-protecting-structure-parameters-ai-models-liang-0xmxc/>, Apr. 2025
>
> ---
>
> **Comment 2**:
> It remains unclear whether SELF is robust to parameter perturbations in non-attention layers (e.g., MLP, LayerNorm). The authors could explore whether these components should be incorporated into fingerprint construction.
>
> **Response**:
> We appreciate the reviewer's suggestion to investigate the robustness of non-attention parameters. Parameter perturbations in non-attention layers do not affect our results. As detailed in Sec. 4.2, we derive fingerprints exclusively from $W_Q$, $W_K$, $W_V$, and $W_O$ in attention blocks. Consequently, perturbations in components such as MLPs and LayerNorm do not alter the extracted fingerprint, leaving the similarity detection results unaffected.
>
> The rational for excluding other components in our method include: (1) Low-dimensionality (for LayerNorm): Attention matrices encodes rich, high-dimensional features (e.g.,$W_Q$, $W_K$ account for $>$30\% of Llama2-7B’s parameters), while LayerNorm has a much smaller parameter size (e.g., 0.1\% of Llama2-7B’s total parameters) and outputs vectors, which limit its utility for SVD- and EVD-based fingerprint extractions. (2) Architectural non-universality (for MLP): Attention blocks are universal across transformer variants, while other blocks (e.g. MLPs in MoE models) exhibit structural variability that could undermines consistency.
>
> Accordingly, we have added a discussion in Sec. 4.2 elaborating these points. Our experiments demonstrate that attention-based fingerprints are sufficient for accurate IP detection. Future work could explore the role of non-attention layers in fingerprint construction for models where these components dominate.

---

> > ### Author Response · Authors · 2025-11-28
> > **Response to Reviewer QAmV (2)**
> >
> > **Comment 3**:
> > It is uncertain how the method adapts to structural variations in the model, such as changes in the number of layers (either adding or removing layers).
> >
> > **Response**:
> > Thanks for the reviewer's comment. To evaluate SELF's robustness against structural variations, we discuss the following scenarios:
> >
> > (1) Adding layers: The scenario of "adding layers" for LLM typically arises in methods like Adapter-based fine-tuning [ref2]. In these methods, lightweight modules (e.g., adapter) are inserted between the existing Transformer blocks (e.g., after the attention or feed-forward blocks). Crucially, the original pre-trained parameters of the backbone model are frozen during this process to prevent catastrophic forgetting. Since SELF extracts fingerprints directly from the Self-Attention weights ($W_Q, W_K, W_V, W_O$) of the original front layers, and these specific weight matrices remain static in such fine-tuning, the fingerprint extraction process remains completely unaffected by the addition of these external layers.
> >
> > (2) Removing layers: layer-pruning is a commonly used technique that may lead to reduced number of layers. As discussed in [ref3], layer-pruning techniques often target deeper (close-to-output) layers in order to preserve original model performance. Instead, our method exploits the first 8 layers (close-to-input) for fingerprint extraction, which would not be affected by the layer pruning techniques; To further validate this, we conducted an ablation study evaluating SELF's performance (by fingerprint margin) when constructing fingerprints using different number ($N_\mathcal{F}$) of early layers. Positive fingerprint margins shown in the below table (Also reported in Table 15 in Appendix F) indicates that even when the model is pruned to fewer than 8 layers, our method still maintains strong discriminability, demonstrating its resilience to such attacks.
> >
> > |Layers: | First 2 Layers | First 4 Layers | First 6 Layers| First 8 Layers |
> > | :--------------| :--------------| :--------------| :-------------| :--------------|
> > |Fingerprint Margin:| 0.0911         | 0.2080         | 0.2794        | 0.3319         |
> >
> > (3) Structural pruning: Structural pruning is a technique used to reduce the size and complexity of neural networks by removing entire structural components (e.g., neurons, attention heads, layers, or blocks). SELF can also maintain excellent performance for structural pruning conditions. As reported in Table 5 of Appendix C, SELF outputs high similarity for all 6 structured pruned versions of Llama2-7B, indicating successful detection of related models. Moreover, we also evaluate SELF under different structural pruning ratios. As shown in Fig. 6c of Sec. 5.2, the results show that SELF with SimNet is able to reliably identify significantly pruned models (ratio=0.25, PPL-PTB=60.92), confirming our method's robustness against structural pruning attacks.
> >
> > [ref2] Houlsby, Neil, et al. *Parameter-efficient transfer learning for NLP*. International conference on machine learning. PMLR, 2019.
> >
> > [ref3] Gromov A, Tirumala K, Shapourian H, et al. *The Unreasonable Ineffectiveness of the Deeper Layers*. The Thirteenth International Conference on Learning Representations, 2025.

---

> > > ### Author Response · Authors · 2025-11-28
> > > **Response to Reviewer QAmV (3)**
> > >
> > > **Comment 4**:
> > > Possible data leakage: Although the training and testing sets are nominally separated, the augmented samples used in training are highly similar to the original models, and the evaluation extensively involves these same or similar models and their fingerprints. This raises concerns about potential overlap in the feature distribution between training and test sets, possibly leading to overfitting. Furthermore, the generalization ability of SELF to unseen models has not been convincingly demonstrated. The authors are encouraged to clearly demarcate the training and testing model spaces and to include cross-family generalization experiments involving structurally diverse models to enhance credibility.
> > >
> > > **Response**:
> > > We appreciate the reviewer’s rigorous examination regarding data leakage and generalization. We address these concerns by clarifying our few-shot training design and providing cross-family generalization evidence.
> > >
> > > (1) Few-shot training: As shown in Appendix E.2, SimNet is trained with only a subset of models, including the target model, one of its variants, and three unrelated models. This few-shot training strategy minimizes memorization risk.
> > >
> > > (2) Generalization to unseen models: **SELF has been rigorously evaluated on a diverse of unseen models, including both related variants and unrelated models from different model families and structures**. Despite limited training data, Our experiments (as reported in the manuscript) demonstrate that SELF achieves high accuracy in distinguishing **unseen** related and unrelated models, confirming it learns a robust distance metric rather than memorizing specific fingerprints. For example, we use Llama2-7B as the target model and the models used for its SimNet training are marked with * in the table below. The SimNet scores show that for those unseen models (models not marked with *), SELF exhibits high discriminability between related and unrelated models ($>$ 0.9 for unseen related models and $<$0.3 for unseen unrelated models).  It demonstrates that unseen models can still be accurately identified with high discriminability, even after heavy fine-tuning (CodeLlama-7B, Llemma-7B) or structural pruning (Sheared-Llama-2.7B). The experiments and analysis can be found in Sec. 5.1.
> > >
> > > | **Model (Unrelated)** | **SimNet Score** | **Model (Related)** | **SimNet Score** |
> > > | :--- | :--- | :--- | :--- |
> > > | Mistral-7B-V0.3* | 0.0050 | **Fine-tuned Variants** | |
> > > | Qwen1.5-7B | 0.2902 | Llama-2-7B-Chat* | 0.9950 |
> > > | Baichuan2-7B | 0.1862 | CodeLlama-7B | 0.9641 |
> > > | InternLM2.5-7B* | 0.0050 | Llemma-7B | 0.9699 |
> > > | GPT2-Large | 0.2372 | **Pruned Variants** | |
> > > | Cerebras-GPT-1.3B | 0.2874 | Sheared-Llama-2.7B | 0.9917 |
> > > | ChatGLM2-6B | 0.0057 | SparseLlama-2-7B | 0.9941 |
> > > | OPT-6.7B | 0.0009 | **Quantized Variants** | |
> > > | Pythia-6.9B | 0.0002 | Llama2-7B-4bit | 0.9950 |
> > > | MPT-7B* | 0.0050 | Llama2-7B-8bit | 0.9950 |
> > >
> > > (3) Cross-Family Generalization: To demonstrate broader applicability, we evaluate SELF on Qwen2.5-7B, a structurally distinct model family from previous Llama2-7B. As shown in the table below, our method maintains high-confidence discrimination  ($>$0.9 for related models and $<$0.1 for unrelated models), confirming effective generalization across architectures. These results have been incorporated into Sec. 5.1 in the manuscript.
> > >
> > > | **Model (Unrelated)** | **SimNet Score** | **Model (Related)** | **SimNet Score** |
> > > | :--- | :--- | :--- | :--- |
> > > | Mistral-7B-V0.3 | 0.0050 | **Fine-tuned Variants** | |
> > > | Llama2-7B | 0.0020 | Qwen2.5-7B-Instruct | 0.9950 |
> > > | Baichuan2-7B | 0.0009 | Qwen2.5-Math-7B | 0.9979 |
> > > | InternLM2.5-7B | 0.0139 | Qwen2.5-Coder-7B | 0.9910 |
> > > | GPT2-Large | 0.0050 | TableGPT2-7B | 0.9944 |
> > > | Cerebras-GPT-1.3B | 0.0011 | Qwen2.5-7B-Medicine | 0.9950 |
> > > | ChatGLM2-6B | 0.0091 | Qwen2.5-7B-abliterated-v2 | 0.9950 |
> > > | OPT-6.7B | 0.0005 | **Quantized Variants** | |
> > > | Pythia-6.9B | 0.0156 | Qwen2.5-7B-4bit | 0.9950 |
> > > | MPT-7B | 0.0050 | Qwen2.5-7B-8bit | 0.9950 |

---

> > > > ### Author Response · Authors · 2025-11-28
> > > > **Response to Reviewer QAmV (4)**
> > > >
> > > > **Comment 5**:
> > > > The method may be vulnerable to model distillation-based stealing attacks. In particular, results on FuseLLM-7B in Appendix F suggest that SELF struggles to maintain robustness in such scenarios.
> > > >
> > > > **Response**:
> > > > We appreciate the reviewer's insightful observation. Distillation-based stealing (e.g., FuseLLM-7B) indeed poses a fundamental challenge for weight-based fingerprinting methods, as they mimic output distributions rather than copying/modifying weights directly. Our analysis reveals this:
> > > > (1) FuseLLM-7B's relation to OpenLlama-2-7B/MPT-7B is undetectable because it only uses their output distributions, not their weights. However, SELF can still detect FuseLLM-7B's realtion to its base model (Llama2-7B), as weight inheritance persists. While REEF achieves higher similarity scores ($>0.62$) for these cases, Table 8 and 9 show these values overlap with unrelated models, risking misclassification.
> > > > (2) This limitation is inherent to weight-based approaches. To the best of our knowledge, there is currently no weight-based fingerprinting method capable of accurately identifying distillation-based stealing. In the future work, we will explore hybrid fingerprints (e.g., combining weight-based fingerprints with activation-based watermarks to trace distillation) to address this gap. This discussion of limitation and future work has been appended to the last paragraph of Appendix C.

---

### Official Review · Reviewer_63A4 · 2025-10-31

**Soundness:** 2
**Presentation:** 2
**Contribution:** 2
**Rating:** 2
**Confidence:** 5

**Summary:**

The paper proposes SELF, a weight-based LLM fingerprinting method that defends against both permutation attacks and linear-mapping attacks by exploiting matrix invariances—using singular values (invariant under row/column permutations) and eigenvalues (invariant under similarity transformations) to build robust fingerprints.

**Strengths:**

1. The paper's mathematical formulation is clear. The choice to use singular values and eigenvalues to counter specific transformation attacks (permutation and linear mapping) is well-justified with solid and reliable theoretical derivations.

2. The proposed methodology demonstrates practical feasibility. The entire pipeline, from fingerprint extraction to similarity comparison, is well-defined.

**Weaknesses:**

1. The experimental validation for related models feels insufficient for some categories. For instance, while the paper claims to effectively identify pruned, quantized, False Claim Attack, and merged models, the main results table only lists the similarity score for a single model or none in each of these classes. Broadening the evaluation to include more diverse examples for each modification type would provide stronger evidence for the method's generalizability.

2. The experiments are centered on the Llama and Llama2 families. The study would be more comprehensive and timely if it included an evaluation of newer, state-of-the-art models. For example, models such as Qwen2.5-7B and its many variants are now widely available and would serve as an excellent test case for the robustness and scalability of the SELF method.

3. The paper does not propose a clear or principled method for setting a suitable discrimination threshold for similarity scores. This becomes particularly problematic in cases like those shown in Appendix Table 11, where the scores for heavily fine-tuned models like Codellama-7B (500B Tokens) and Llemma-7B (700B Tokens) are significantly lower than other related models, yet are still considered correct detections. Without a defined threshold, it is difficult to assess the method's reliability in these challenging, borderline scenarios.

**Questions:**

1. All reported results are based on raw similarity scores. How do SELF, REEF, and ICS compare when each method applies its own paper-defined decision threshold to convert similarity into binary judgments (related vs. unrelated)? Please report thresholded metrics such as accuracy, precision/recall/F1, and confusion matrices for a fair, apples-to-apples comparison.

2. For routine model checks (e.g., those in Table 6 and the additional models you included), how does a simple classifier based on Fingerprint Distance (with a principled threshold or ROC/AUC analysis) perform relative to training a SimNet? Quantitatively, how large is the accuracy gap, and in which settings (fine-tuning, pruning, quantization) does SimNet provide the most gain?

---

> ### Author Response · Authors · 2025-11-28
> **Response to Reviewer 63A4**
>
> We thank the reviewer for the valuable feedback. We have grouped the identified weakness and questions into the following comments for clarity and response.
>
> **Comment 1**:
> The experimental validation for related models feels insufficient for some categories. For instance, while the paper claims to effectively identify pruned, quantized, False Claim Attack, and merged models, the main results table only lists the similarity score for a single model or none in each of these classes. Broadening the evaluation to include more diverse examples for each modification type would provide stronger evidence for the method's generalizability.
>
> **Response**:
> We appreciate the reviewer's feedback on enhancing the diversity of our validation. To strengthen the evidence for generalizability, we have expanded our experiments as follows:
>
> (1) Pruned models: We supplement the experiments on structured pruning at varying ratios in Sec. 5.2.2 (Fig. 3c), which confirm the robustness of our method across pruning intensities. Additionally, Appendix C (Table 5 and Table 6) evaluates 6 structured and 3 unstructured pruning methods, covering diverse pruning strategies.
>
> (2) Quantized models: We expand our evaluation on quantization attacks to include multiple precision levels (4-bit and 8-bit) in Table 1 (Sec. 5.1), spanning common precision levels.
>
> (3) False claim attack: Since false claim attacks are executed by manipulating the detection input, our input-free weight-based fingerprinting method inherently circumvents this type of attacks.
>
> (4) Merged models: We conducted the evaluation and analysis on both weight-merging and distribution merging in Table 7 (Appendix C), covering common types of model merging schemes.
>
> ---
>
> **Comment 2**:
> The experiments are centered on the Llama and Llama2 families. The study would be more comprehensive and timely if it included an evaluation of newer, state-of-the-art models. For example, models such as Qwen2.5-7B and its many variants are now widely available and would serve as an excellent test case for the robustness and scalability of the SELF method.
>
> **Response**:
> We thank the reviewer for the valuable suggestion. Accordingly, we have replaced Llama-7B with Qwen2.5-7B as the target model in Sec. 5.1. The results (Table 1, also summarized in the table below) confirm that our method achieves high distinguishability between related and unrelated Qwen2.5-7B variants, with similarity scores showing a clear separation ($>$0.9 for realted models vs. $<$0.1 for unrelated models). This underscores the robustness and scalability of SELF across state-of-the-art architectures.
>
> | **Model (Unrelated)** | **SimNet Score** | **Model (Related)** | **SimNet Score** |
> | :--- | :--- | :--- | :--- |
> | Mistral-7B-V0.3 | 0.0050 | **Fine-tuned Variants** | |
> | Llama2-7B | 0.0020 | Qwen2.5-7B-Instruct | 0.9950 |
> | Baichuan2-7B | 0.0009 | Qwen2.5-Math-7B | 0.9979 |
> | InternLM2.5-7B | 0.0139 | Qwen2.5-Coder-7B | 0.9910 |
> | GPT2-Large | 0.0050 | TableGPT2-7B | 0.9944 |
> | Cerebras-GPT-1.3B | 0.0011 | Qwen2.5-7B-Medicine | 0.9950 |
> | ChatGLM2-6B | 0.0091 | Qwen2.5-7B-abliterated-v2 | 0.9950 |
> | OPT-6.7B | 0.0005 | **Quantized Variants** | |
> | Pythia-6.9B | 0.0156 | Qwen2.5-7B-4bit | 0.9950 |
> | MPT-7B | 0.0050 | Qwen2.5-7B-8bit | 0.9950 |

---

> ### Author Response · Authors · 2025-11-28
> **Response to Reviewer 63A4 (2)**
>
> **Comment 3**:
> The paper does not propose a clear or principled method for setting a suitable discrimination threshold for similarity scores. This becomes particularly problematic in cases like those shown in Appendix Table 11, where the scores for heavily fine-tuned models like Codellama-7B (500B Tokens) and Llemma-7B (700B Tokens) are significantly lower than other related models, yet are still considered correct detections. Without a defined threshold, it is difficult to assess the method's reliability in these challenging, borderline scenarios.
>
> **Response**: We thank the reviewer for this insightful comment. To eliminate borderline scenarios (e.g., heavily fine-tuned models like Codellama-7B (500B Tokens) and Llemma-7B (700B Tokens)), we optimized our fingerprint extraction method by (1) Expanding fingerprint sources: incorporating $W_V$ and $W_O$ (in addition to $W_Q$ and $W_K$) to capture richer model invariants and (2) Refining the calculation of the eigenvalue invariant matrix. The updated singular value invariant matrices are extracted as:
>
> $$
>             X_\sigma = W_Q W_K^T \in \mathbb{R}^{d_{\text{model}} \times d_{\text{model}}},
>             Y_\sigma = W_V W_O \in \mathbb{R}^{d_{\text{model}} \times d_{\text{model}}}
> $$
>
> And the updated eigenvalue invariant matrices are extracted as:
>
> $$
>             X_\lambda = W_Q^T W_K \in \mathbb{R}^{d \times d},
>             Y_\lambda = W_O W_V \in \mathbb{R}^{d \times d}
> $$
> These changes ensure consistent high similarity scores ($>0.9$) for related models, even in challenging cases, and widen the margin between related/unrelated models. This simplifies threshold selection and enhances reliability.
> Accordingly, we have updated Sec. 4.2 with the optimized fingerprint extraction and updated the experimental results in all tables.
>
> ---
>
> **Comment 4**:
> All reported results are based on raw similarity scores. How do SELF, REEF, and ICS compare when each method applies its own paper-defined decision threshold to convert similarity into binary judgments (related vs. unrelated)? Please report threshold metrics such as accuracy, precision/recall/F1, and confusion matrices for a fair, apples-to-apples comparison.
>
> **Response**:
> We appreciate the reviewer’s suggestion to include threshold-based metrics.
> Regarding the specific thresholds for baseline methods (REEF, ICS), we reviewed their original publications and found that none explicitly define a fixed classification threshold. However, their reported results demonstrate a separation margin that justifies a universal threshold. Specifically:
>
> *ICS* reports similarity scores peaking at ~0.46 for unrelated models, while related models score significantly higher (e.g., $>$0.75 in their metric).
>
> *REEF* shows a clear boundary where unrelated models score below around 0.2 and related models score above 0.62.
>
> Since a normalized threshold of 0.5 successfully separates positive and negative samples in the original contexts of these methods (achieving perfect discrimination), we adopted 0.5 as the uniform decision boundary for the comparative analysis in Appendix C. We have also explicitly marked false judgments in gray to facilitate a detailed inspection of the results. Table 4$\sim$9 in Appendix C can clearly show that our method outperforms ICS and matches REEF.
>
> ---
>
> **Comment 5**:
> For routine model checks (e.g., those in Table 6 and the additional models you included), how does a simple classifier based on Fingerprint Distance (with a principled threshold or ROC/AUC analysis) perform relative to training a SimNet? Quantitatively, how large is the accuracy gap, and in which settings (fine-tuning, pruning, quantization) does SimNet provide the most gain?
>
> **Response**:
> Thanks for the reviewer’s comment. We conducted the evaluation using fingerprint distance in Appendix D.
> (1) Routine checks (Table 1 vs. Table 10): fingerprint distance achieves comparable accuracy to SimNet for routine models (e.g., unmodified or publicly available variants); (2) Adversarial settings: distance-based detection fails in pruning attacks (Figure 6c of Appendix E.5), and shows a higher growing trend of flipping the verification result with the increase of fine-tuning steps (Figure 6a and 6b of Appendix E.5).
>
> These results confirm that SimNet's training overhead is justified for adversarial scenarios, while distance suffices for routine checks. We've added this analysis to Appendix E.5 for clarity.

---

### Official Review · Reviewer_f8CQ · 2025-11-01

**Soundness:** 3
**Presentation:** 3
**Contribution:** 2
**Rating:** 6
**Confidence:** 5

**Summary:**

This paper proposes SELF, a novel intrinsic weight-based fingerprinting scheme that eliminates dependency on input and inherently resists false claims. Experimental results demonstrate that SELF maintains high IP infringement detection accuracy while showing strong robustness against various downstream modifications. The proposed method is simple and effective, but there are deficiencies in its experimental design and completeness.

**Strengths:**

- The paper is clearly written and highly readable.
- The method is supported by theoretical guarantees.
- The proposed method is computationally efficient, and its effectiveness and robustness are demonstrated experimentally.

**Weaknesses:**

- Due to its reliance on model weights, the method is not applicable to closed-source models. This limits its practical application value, especially given that modern AI applications are primarily offered as services.
- Missing details in the experimental setup:
  - The rationale for choosing attention blocks over other modules (and a comparison of the results).
  - The reason for selecting the first N layers, as opposed to using all layers or a different set of M layers.
  - The source of the SimNet training data, especially considering the potential imbalance between "unrelated" and "target" models. It is also unclear if training SimNet with randomly generated fingerprints against the target model's fingerprint would be similarly effective.
- Unfair comparisons:
  - The comparison with other methods is limited to fingerprint size. However, the SELF method introduces a SimNet, which requires separate training for each target model, representing a significant overhead that is not accounted for.
  - The numerical comparison in Table 4 is not objective, as the values being compared are derived from different methodological systems.
- The models used (e.g., Llama 7B as the suspect model) are relatively old. It is uncertain whether the method is applicable to current mainstream Mixture of Experts (MoE) architectures or how it performs on other model families and larger-scale models.
- There appears to be a typo in Lines 909-911 regarding the definition of "fingerprint margin." It seems it should be defined by the minimum value for related models and the maximum value for unrelated models.

**Questions:**

- As shown in Appendix D, L2 distance appears to be an effective metric. What is the necessity of training a SimNet?
- The method extracts fingerprints from the first 8 layers. Would this method fail if the model undergoes layer-pruning techniques?

---

> ### Author Response · Authors · 2025-11-28
> **Response to Reviwer f8CQ**
>
> We thank the reviewer for the valuable feedback. We have grouped the identified weakness and questions into the following comments for clarity and response.
>
> **Comment 1**: Due to its reliance on model weights, the method is not applicable to closed-source models. This limits its practical application value, especially given that modern AI applications are primarily offered as services.
>
> **Response**: Thanks for the reviewer's insightful comment. Our work focuses on IP forensic detection, where a trusted third party (TTA) can obtain temporary white-box access to both original and suspect models for model ownership investigation. This aligns with legal workflows, as exemplified by the real-world case: Douyin (TikTok) v. B612 (2025) [ref1]. In this landmark case, the Beijing IP court (acting as the TTA) ruled in favor of Douyin after identifying structural and parametric similarities between Douyin's AI transformation model and B612's infringing model in its Kaji APP. Our method can provide accurate, robust, and efficient infringement detection for such scenarios.
>
> While white-box access restricts applicability to auditable or open-source models, our method offers critical advantages over black-box alternatives: (1) zero procedural overhead: our method does not require any retraining or model modifications, offering a lightweight and flexible solution which enables the IP protection of models that have been deployed in the markets; (2) enhanced security: our proposed fingerprints are constructed without relying on data input, eliminating the risk of forging fingerprint by crafting inputs, thus avoid false claim attacks.
>
> Accordingly, we have revised our threat model in Sec. 4.1 to clearly explain the TTA's role and forensic workflow of our proposed work.
>
> [ref1]: China's first case on protecting structure and parameters of AI models: Douyin v. B612 Kaji. <https://www.linkedin.com/pulse/china-first-case-protecting-structure-parameters-ai-models-liang-0xmxc/>, Apr. 2025

---

> ### Author Response · Authors · 2025-11-28
> **Response to Reviwer f8CQ (2)**
>
> **Comment 2**: Missing details in the experimental setup: 1) The rationale for choosing attention blocks over other modules (and a comparison of the results).  2) The reason for selecting the first N layers, as opposed to using all layers or a different set of M layers. 3) The source of the SimNet training data, especially considering the potential imbalance between "unrelated" and "target" models. It is also unclear if training SimNet with randomly generated fingerprints against the target model's fingerprint would be similarly effective."
>
> **Response**: We thank the reviewer for these valuable suggestions. We have provided a point-to-point explanation on each of the three experimental setups below.
>
> **Rational for choosing attentional blocks**: Attention blocks are selected due to their **high-dimensional features** and **architectural universality**. Specifically,
>
> (1) High-dimensionality: Attention matrices encodes rich, high-dimensional features (e.g., $W_Q$, $W_K$ account for $>$30\% of Llama2-7B's parameters), while LayerNorm's parameter scarcity (e.g., 0.1\% of Llama2-7B's total parameters) and vector output limit its utility for SVD- and EVD-based fingerprint extraction.
>
> (2) Architectural generality: Attention blocks are universal across transformer variants, while other blocks such as MLPs (e.g., in MoE models) exhibit structural variability that could undermines consistency.
>
> Our experiments presented in the paper confirms that attention weights yield high discriminative power, achieving clear separation between related and unrelated models. This further justifies our design choice. Accordingly, we have added discussion on rational for choosing attentional blocks in Sec. 4.2.
>
> **Reason for selecting the first N layers**: We select the first N layers for the following reasons: (a) Stability: The front layer weights tend to remain relatively stable even when the model undergoes subsequent modifications or fine-tuning, as studied by the previous works [ref2][ref3]. (b) Scalability and Generality: Fixing the selection to the first N layers ensures our method is broadly applicable across models with varying numbers of layers. Moreover, most existing LLMs have more than 8 layers, making this choice both practical and widely compatible.
>
> We conducted ablation study about the impact of *layer selection* and *layer numbers (Varying $N_{\mathcal{F}}$)* on IP detection performance. We use fingerprint margin ($fm=min\_{fingerprint}^{\text{unrelated}} - max\_{fingerprint}^\text{related}$) to evaluate the performance. A higher margin indicates higher discriminability. The following empirically proved that the fingerprint extracted from the attention weights of the first 8 layers exhibit the best discriminability of IP, and optimal discriminability is achieved with the selected layer number configuration ($N_\mathcal{F}=8$). The experimental results and discussions can also be found in Appendix F.
>
> Layers: | First 2 Layers | First 4 Layers | First 6 Layers| First 8 Layers | Middle 8 Layers | Last 8 Layers |
> | :--------------| :--------------| :--------------| :-------------| :--------------| :-------------- | :-------------|
> | fm:  | 0.0911         | 0.2080         | 0.2794        | 0.3319         | -0.1380         | -0.1263       |
>
>
>
> [ref2] Yosinski, Jason, et al. *How transferable are features in deep neural networks?*. Advances in neural information processing systems 27, 2014
>
> [ref3] Yue Zheng, Si Wang, and Chip-Hong Chang. *A dnn fingerprint for non-repudiable model ownership
> identification and piracy detection*. IEEE Transactions on Information Forensics and Security, 17:
> 2977–2989, 2022
>
> **Source of the SimNet training data**: We have listed the SimNet training data for all target models evaluated in the paper in Appendix E.2. Generally speaking, the training data for target model's SimNet includes: target model and one of its related variants, and three unrelated models, as well as their respective augmented versions (12 per model). Therefore, each target model has 26 related samples and 39 unrelated samples, leading to comparable number of the “unrelated” and “targeted” models used for SimNet training.
>
> We thank the reviewer's suggestion of ``using randomly generated fingerprints`` for SimNet training. Correspondingly, we conducted the following experiment: Using Llama2-7B as the target model, we randomly generated three sets of $W_Q, W_K, W_V, W_O \in 4096 \times 4096$ from a standard distribution and follow the same fingerprint extraction and SimNet training strategy. The results presented in the below table show that using randomly generated fingerprints for training cannot distinguish related and unrelated models:

---

> > ### Author Response · Authors · 2025-11-28
> > **Response to Reviwer f8CQ (3)**
> >
> > | **Llama-2-7b Unrelated** | **SimNet Score** | **Llama-2-7b Related** | **SimNet Score** |
> > |:-------------------------|:------|:-----------------------|:------|
> > | Mistral-7B-v0.3          | 0.99892277 | Llama-2-7B-chat   | 0.99489248 |
> > | Qwen1.5-7B               | 0.99528414 | CodeLlama-7B       | 0.99895501 |
> > | Baichuan2-7B-Base        | 0.99999535 | vicuna-7B-v1.5         | 0.99504781 |
> > | internlm2.5-7B           | 0.99839109 | WizardMath-7B-V1.0     | 0.99501133 |
> > | gpt2-large               | 0.99887294 | llemma-7B              | 0.99945301 |
> > | Cerebras-GPT-1.3B        | 0.99998367 | Llama2-7B-Finance      | 0.99488467 |
> > | chatglm2-6B              | 0.99999559 | Chinese-Llama-2-7B     | 0.99526227 |
> > | opt-6.7B                 | 1.00000000 | Sheared-LLaMA-2.7B     | 0.99804819 |
> > | pythia-6.9B              | 0.09019652 | SparseLlama-2-7B | 0.99570447 |
> >
> > ---
> >
> > **Comment 3**: Unfair comparisons: 1) The comparison with other methods is limited to fingerprint size. However, the SELF method introduces a SimNet, which requires separate training for each target model, representing a significant overhead that is not accounted for. 2) The numerical comparison in Table 4 is not objective, as the values being compared are derived from different methodological systems.
> >
> > **Response**: We thank the reviewer for these valuable suggestions. We have provided a point-to-point explanation below.
> >
> > (1) SimNet Overhead: We appreciate the reviewer's observations. While separate training of SimNet is required, our method only requires **one-time** training per target model, and SimNet can be reused for all subsequent detections, which ensures the overhead is amortized over time. Furthermore, SimNet is relatively lightweight in terms of architecture, training data, training resource as well as inference time. Specifically, we adopt ResNet with 5 residual blocks as the architecture of SimNet and use only 65 fingerprints for training. Training SimNet only takes about 2.5 minutes using a single RTX4090 GPU, and one inference takes less than 0.1 seconds. This minor overhead, however, enables superior performance of our proposed method. As demonstrated in Appendix C, SimNet ensures a superior distinguishability between related and unrelated models over existing methods. Correspondingly, we have clarify these points in Appendix E.4 to ensure the cost-benefit balance is transparent.
> >
> > (2) Numerical Comparison: We thank for the reviewer's comment. While REEF and our method employ different methodologies, this comparison (Table 8 in the updated version) objectively highlighted the **input-dependence instability (marked by gray) of REEF**, i.e., REEF's similarity scores for unrelated models could be abnormally high for some datasets (including PKU-SafeRLFH, ToxiGen and SST). This presents a critical limitation that risks inconsistent judgments and false claim attacks. As demonstrated in Table 9, for the above three datasets, REEF's similarity between the distribution merged FuseLLM-7B and its source models, OpenLlama-2-7B and MPT-7B, could be lower than those unrelated models in Table 8, leading them to be detected as unrelated despite their relatedness. These overlap confirms that REEF lacks adequate discriminative power using these datasets. In contrast, our input-free design achieves stable, low similarity scores for all unrelated models, eliminating sampling bias and adversarial manipulation entirely. This underscores our method's reliability for real-word IP protection, where consistency is important.

---

> ### Author Response · Authors · 2025-11-28
> **Response to Reviwer f8CQ (4)**
>
> **Comment 4**:
> The models used (e.g., Llama 7B as the suspect model) are relatively old. It is uncertain whether the method is applicable to current mainstream Mixture of Experts (MoE) architectures or how it performs on other model families and larger-scale models.
>
> **Response**:
> We thank the reviewer for this comment. To evaluate our method's effectiveness on other model families, we have replaced Llama-7B with Qwen2.5-7B as the target model in Sec. 5.1, and the similarity results (Table 1 in the manuscript) are reproduced below. The results show that the related and unrelated models of Qwen2.5-7B can still be accurately distinguished using our method.
>
> | **Model (Unrelated)** | **SimNet Score** | **Model (Related)** | **SimNet Score** |
> | :--- | :--- | :--- | :--- |
> | Mistral-7B-V0.3 | 0.0050 | **Fine-tuned Variants** | |
> | Llama2-7B | 0.0020 | Qwen2.5-7B-Instruct | 0.9950 |
> | Baichuan2-7B | 0.0009 | Qwen2.5-Math-7B | 0.9979 |
> | InternLM2.5-7B | 0.0139 | Qwen2.5-Coder-7B | 0.9910 |
> | GPT2-Large | 0.0050 | TableGPT2-7B | 0.9944 |
> | Cerebras-GPT-1.3B | 0.0011 | Qwen2.5-7B-Medicine | 0.9950 |
> | ChatGLM2-6B | 0.0091 | Qwen2.5-7B-abliterated-v2 | 0.9950 |
> | OPT-6.7B | 0.0005 | **Quantized Variants** | |
> | Pythia-6.9B | 0.0156 | Qwen2.5-7B-4bit | 0.9950 |
> | MPT-7B | 0.0050 | Qwen2.5-7B-8bit | 0.9950 |
>
> As for the MOE architecture, our method is applicable because it exclusively targets the fundamental attention module, which is an invariant component across all standard Transformer architectures, including MoE models. Modern MoE architectures primarily replace the dense FNN with a sparse Expert Layer, leaving the attention module mathematically and structurally intact. By focusing on the attention weights, our method maintains generality and effectiveness.
>
> We acknowledged that due to resource constraints, we were unable to conduct experiments for larger-scale models. However, the existing experiments confirm the effectiveness of our method's core concept, and future work will include validation on larger models as resource permits.
>
> ---
>
> **Comment 5**: There appears to be a typo in Lines 909-911 regarding the definition of "fingerprint margin." It seems it should be defined by the minimum value for related models and the maximum value for unrelated models".
>
> **Response**: Thank you for your attention to detail. This definition is intentional instead of a typo: to ensure robust discriminability, we evaluate the **worst-case** scenarios--requiring that the *minimum* fingerprint distance between unrelated models exceeds the *maximum* fingerprint distance between related models. Note that a larger distance indicates a less similarity, thus the higher possibility of being unrelated models. This **fingerprint margin** quantifies the distinctiveness of our method, where a larger margin indicates stronger separation between related and unrelated models. We appreciate the opportunity to clarify this point.
>
> ---
>
> **Comment 6**:
> As shown in Appendix D, L2 distance appears to be an effective metric. What is the necessity of training a SimNet?
>
> **Response**:
> Thanks for the reviewer’s comment. While L2 distance achieves comparable results on standard models (as shown in Appendix D), SimNet is critical for robustness IP detection against pruning attacks. As demonstrated in Appendix E.5, distance-based detection fails against structured pruning attacks--a common evasion tactic in model theft. SimNet, however, maintains stable performance by learning invariant features beyond straightforward distance metrics. This ensures reliable IP detection in real world.
>
> ---
>
> **Comment 7**:
> The method extracts fingerprints from the first 8 layers. Would this method fail if the model undergoes layer-pruning techniques?
>
> **Response**:
> Thanks for the reviewer’s comment. As discussed in [ref4], layer-pruning techniques often target deeper (close-to-output) layers in order to preserve original model performance. Instead, our method exploits the first 8 layers (close-to-input) for fingerprint extraction, which would not be affected by the layer pruning techniques. To further validate this, we conducted an ablation study evaluating fingerprint margin when constructing fingerprints using different number ($N_\mathcal{F}$) of early layers (please check response to comment 2). The results show that even when the model is pruned to fewer than 8 layers, our method still maintains strong discriminability, demonstrating its resilience to such attacks.
>
> [ref4] Gromov A, Tirumala K, Shapourian H, et al. *The Unreasonable Ineffectiveness of the Deeper Layers*. The Thirteenth International Conference on Learning Representations, 2025. where adversaries actively obscure provenance.

---

### Author Response · Authors · 2025-11-28
**Response to AC and all Reviewers**

We sincerely thank the reviewers for their constructive comments and careful reading of our paper: **SELF: A Robust Singular Value and Eigenvalue Approach for LLM Fingerprinting**.

We proposed SELF, an **efficient, robust, and scalable** fingerprinting method for LLM IP infringement detection, addressing real-world needs as exemplified by China's landmark "Douyin v. B612" case (the first to protect AI model structure and parameters). SELF extracts LLM fingerprints by constructing **singular value and eigenvalue invariant matrices** from model weights and leverages a few-shot trained SimNet to reliably discriminate infringements. Our method is built on rigorous theoretical foundations, validated across cross-family models and architectures, and evaluated under a variety of practical adversarial scenarios including fine-tuning, pruning, quantization, distillation and model merging. Compared to SOTA LLM fingerprinting methods, SELF is resilient to false claim and weight transformation attacks, achieves high discrimination rate among *unseen* related and unrelated models, and operates with a much smaller fingerprint size and lower computation cost.

---

We are grateful that the reviewers recognized our work's clear mathematical formulation, efficient and effective design, well-defined pipeline, and comprehensive empirical validation. We are glad to inform the AC and the reviewers that **we have carefully addressed all concerns and significantly improved the manuscript**. Major changes are highlighted in the submitted PDF and a summary of the major revisions is provided below:

(1) **Clarification on Threat Model (White-Box vs. Black-Box)**: Regarding the concern on application constraints, we clarified our target IP Forensic scenario and justified the real-word applicability and necessity. Correspondingly, we revised Sec. 4.1 to explicitly define the Trusted Third Party role and forensic workflow (See response to Reviewer f83Q's comment 1 and Reviewer QAmV's comment 1).

(2) **Clarification on Terminology Misunderstanding**: Regarding the misunderstanding of key terminologies (Fingerprint margin and False claim attack) in our paper, we elaborated the definition of fingerprint margin, clarified the distinction between false claim attack and ambiguity attack, and justified why our original descriptions are accurate and appropriate (See response to Reviewer f83Q's comment 5 and Reviewer HKUo's comment 4).

(3) **Method Optimization**: Regarding the stability concerns on fine-tuned models, we have optimized our method by incorporating $W_V$ and $W_O$ into the fingerprint extraction process and refining the invariant matrix calculation. This optimization significantly enhances performance (See response to Reviewer 63A4's comment 3).

(4) **Demonstration of Cross-Family Generalization**: Regarding the concerns about limited model family scope, we have replaced Llama-7B with **Qwen2.5-7B (GQA architecture)** as the target model. This validates our method's effectiveness across different model families and architectures (See response to Reviewer f8CQ's comment 4 and Reviewer 63A4's comment 2).

(5) **Expanded Validation Experiments**: Regarding the scope of validation experiments, we have expanded our analysis to ensure robustness against various attacks. We added experiments on structured pruning at varying ratios (Sec. 5.2.2) and evaluated 6 structured and 3 unstructured pruned models (Appendix C). Furthermore, we included multiple quantization levels (Sec. 5.1) (See response to Reviewer 63A4's comment 1 and Reviewer HKUo's comment 3).

(6) **Necessity of SimNet vs. Fingerprint Distance**: Regarding the necessity of SimNet versus fingerprint distance, We clarified that while fingerprint distance works for routine model checks, SimNet is essential for robustness against adversarial structured pruning (Appendix E.5) (See response to Reviewer f8CQ's comment 6 and Reviewer 63A4's comment 5).

(7) **Clarification on Layer Selection and Layer Pruning**: Regarding the concern on fingerprint layer selection and robustness against layer pruning, we have clarified that our choice of the first 8 layers (close-to-input) is strategic. While layer-pruning typically target deeper layers, early layers usually remain intact. We conducted an ablation study (Appendix F) to validate this, showing that our method maintains discriminability even when models are pruned to $<8$ layers (See response to Reviewer f8CQ's comments 2 and 7, and Reviewer QAmV's comment 3).

(8) **Clarification on Distribution-Based Stealing**: Regarding the challenge of detecting distribution-based model stealing (e.g., FuseLLM), we note that this is a fundamental limitation for all weight-based fingerprinting methods, as merged models learn from output distributions rather than inheriting weights. Even state-of-the-art REEF exhibits ambiguous score overlaps in such cases (Table 8/9) (See response to Reviewer QAmV's comment 5 and Reviewer HKUo's comment 3).

---

### Meta-Review · Area_Chair_UE6c · 2026-01-06

**Summary:**

This paper proposes a robust LLM (Large Language Model) fingerprinting method by extracting singular values and eigenvalues, which exhibits favorable defensive performance against attacks such as linear transformations and perturbations; however, the reviewers’ evaluations of this paper vary significantly, with their main concerns including limited scope of application (as it can only be applied to white-box models), lack of fairness in the experimental comparisons, and incompleteness of the theoretical proofs. The authors provided a detailed response that has partially addressed these concerns, and in my opinion, the authors could further improve this paper by expanding its scope of application, conducting more fair experiments, and refining the theoretical proofs.

**Reviewer Concerns:**

The main concerns including: limited scope of application (as it can only be applied to white-box models), lack of fairness in the experimental comparisons, and incompleteness of the theoretical proofs.

The lack of fairness in the experimental and lack of fairness in the experimental comparisons have been partially addressed.

**Reviewer Scores:**

None

---

### Decision · Program_Chairs · 2026-01-26

Reject